# Continuously processing waste lignin into high-value carbon nanotube fibers

Fuyao Liu[1], Qianqian Wang[1], Gongxun Zhai[1], Hengxue Xiang [1]✉, Jialiang Zhou[1], Chao Jia [1]✉, Liping Zhu[1], Qilin Wu[1] & Meifang Zhu [1]✉

High value utilization of renewable biomass materials is of great significance to the sustainable development of human beings. For example, because biomass contains large amounts of carbon, they are ideal candidates for the preparation of carbon nanotube fibers. However, continuous preparation of such fibers using biomass as carbon source remains a huge challenge due to the complex chemical structure of the precursors. Here, we realize continuous preparation of high-performance carbon nanotube fibers from lignin by solvent dispersion, high-temperature pyrolysis, catalytic synthesis, and assembly. The fibers exhibit a tensile strength of 1.33 GPa and an electrical conductivity of $1.19 \times 10^5$ S m$^{-1}$, superior to that of most biomass-derived carbon materials to date. More importantly, we achieve continuous production rate of 120 m h$^{-1}$. Our preparation method is extendable to other biomass materials and will greatly promote the high value application of biomass in a wide range of fields.

Carbon fibers have been used in many industries such as aerospace, aircraft, automobile, sports, and medical equipment because of their high modulus and strength[1]. The common precursors for the production of carbon fibers include polyacrylonitrile, petroleum asphalt and regenerated cellulose[2]. In recent years, more and more attention has been paid to using biomass resources to produce carbon fibers instead of petroleum-based resources due to concerns about environmental issues and production costs. Among biomass resources, lignin materials can be obtained from agricultural waste and pulp industry waste, showing significant cost advantages[3]. As the second largest natural polymer material in reserves after cellulose, lignin contains a large number of aromatic ring structures, which can be used as a precursor to prepare carbon fibers with a production cost reduction of more than 50% compared with polyacrylonitrile-based carbon fibers[4].

The methods for preparing carbon fibers from lignin include melt spinning[5], dry spinning[6], wet spinning[7], dry jet wet spinning[8], and electrospinning[9]. Lignin can be processed into carbon fibers through multiple steps, including extraction, purification, spinning, stabilization, and carbonization[10]. However, the tensile strength of lignin-based carbon fibers is only one-third that of polyacrylonitrile-based carbon fibers[11,12]. Due to the nonlinear molecular structure and wide molecular weight distribution of lignin, the preparation of carbon fibers from lignin has some problems, such as poor quality of precursor fibers, uneven and large fiber diameter, spinneret blockage, and so on[12,13]. In order to produce high-quality carbon fibers from lignin, some measures have been adopted to improve the spinnability of lignin, including transforming lignin into a linear polymer[14] and mixing it with other resins[15]. However, these complex treatment processes inevitably increase the production cost of lignin-based carbon fibers, and are not conducive to their large-scale preparation.

Carbon nanotube (CNT) fiber is a special carbon fiber material, which is composed of multiple CNTs and exhibits high specific strength[16]. Compared with traditional carbon fibers, CNT fibers have better flexibility, higher electrical conductivity, and thermal conductivity[17]. The preparation methods of CNT fibers include wet spinning[18], array spinning[19], and floating catalyst chemical vapor deposition (FCCVD)[20]. In addition, post-treatment is usually used to improve the mechanical, electrical, and thermal properties of CNT fibers, including solvent/mechanical densification[21], chemical doping[22], metal coating[23], acid treatment[24], and purification[25]. Currently, the most commonly used method for continuously preparing CNT fibers is FCCVD[26]. The carbon sources used in this method are mainly from petroleum fine chemicals, such as methane, ethylene,

[1]State Key Laboratory for Modification of Chemical Fibers and Polymer Materials, College of Materials Science and Engineering, Donghua University, 201620 Shanghai, China. ✉e-mail: hengxuexiang@dhu.edu.cn; jiachao0806@dhu.edu.cn; zmf@dhu.edu.cn

ethanol, toluene and xylene[27]. In order to meet the requirements of low carbon and environmental protection, the preparation of CNTs from biomass with a low carbon footprint as a carbon source has become a research hotspot in this field[28,29]. However, the continuous preparation of CNT fibers from biomass by FCCVD method still presents great challenges.

In this work, we realize the continuous preparation of high-performance CNT fibers from lignin by solvent dispersion, high-temperature pyrolysis, catalytic synthesis, and assembly. The CNT fibers are synthesized by carbon monoxide (CO) and hydrogen ($H_2$) released from decomposed lignin under the catalysis of ferrocene. After post-treatment, the lignin-based CNT fibers are endowed with a tensile strength of 1.33 GPa and electrical conductivity of $1.19 \times 10^5 \, S \, m^{-1}$. In addition, the continuous production of CNT fibers from lignin with a $120 \, m \, h^{-1}$ production rate is achieved. The excellent mechanical strength and electrical conductivity of lignin-based CNT fibers will greatly expand the application field of lignin.

## Results

### Synthesis mechanism of CNT fibers from lignin

Lignin solution was first obtained by dissolving lignin in an appropriate solvent. The lignin solution is heated in a high-temperature furnace, the solvent evaporates instantly, and the lignin is decomposed into monocyclic aromatic hydrocarbons (MAHs). At temperatures above 1300 °C, these MAHs can be further pyrolyzed into $H_2$ and CO. The decomposition products are synthesized into CNTs catalyzed by iron (Fe) particles and promoted by sulfur (S). The synthesized CNTs are further assembled into a sock-like integrate, which is introduced into water for densification and further twisted or rolled to obtain CNT fibers under drafting force (Fig. 1 and Supplementary Movie 1).

In order to realize continuous preparation of CNT fibers using lignin, it is necessary to select appropriate solvents. In addition to good solubility, the simple structure and low cost of the solvents are essential for the large-scale preparation of high-performance CNT fibers. In addition, a synthetic temperature of higher than 1300 °C is needed to decompose lignin into $H_2$ and CO as raw materials of CNT fibers. Our CNT fibers obtained under these conditions have the advantages of high mechanical strength (1.33 GPa), high electrical conductivity ($1.19 \times 10^5 \, S \, m^{-1}$), and high preparation efficiency ($120 \, m \, h^{-1}$). We achieved the continuous synthesis of high-performance CNT fibers from lignin, breaking the limitation that CNT fibers cannot be continuously prepared from biomass (Supplementary Table 1).

### Mechanism analysis for the lignin pyrolysis

The structure of lignin determines its pyrolysis characteristics. The nuclear magnetic resonance (NMR) spectra of lignin were obtained to analyze the structural characteristics. $^{13}C$ NMR can effectively detect the carbon skeleton structure of lignin and provide a comprehensive analysis of the overall structure of lignin. Figure 2a shows the two-dimensional heteronuclear single quantum coherence (2D-HSQC) spectra of lignin. The corresponding connection units and assignments are shown in Supplementary Figs. 1, 2 and Supplementary Table 2. The syringyl S-type units (103.9/6.7 ppm) and guaiacyl G-type units (110.8/6.97 ppm, 114.5/6.7 ppm, 119.0/6.78 ppm) were identified, β-β resinol (53.5/3.07 ppm, 71.2/3.82–4.18 ppm, 84.8/4.66 ppm) and β-O-4 unit (59.9/3.35–3.80 ppm, 71.8/4.86 ppm) appeared in the HSQC spectra of lignin[30,31]. The signal peaks at 121.03, 115.7 and 111.96 ppm in the $^{13}C$ NMR spectrum correspond to C6, C5 and C2 positions in guaiacyl G-type structure, and the signal peaks at 152.91, 134.65, and 105.4 ppm correspond to the C3/C5, C1 and C2/C6 positions in syringyl S-type structure. The signal peaks at 72.13 ppm and 63.68/59.76 ppm can be assigned to the Cα and Cγ in β-O-4, respectively[31].

The heteronuclear multiple bond coherence (HMBC) spectra and the main linkages are shown in Supplementary Figs. 3 and 4. Hα (H4.76) was found to be connected with the HMBC of Cβ′ and Cγ′, whereas Hγ (H4.10) was discovered to be correlated with the HMBC of Cα, Cβ, Cα′and Cβ′, indicating the presence of β–β bonds in lignin. Furthermore, Hα (H4.63) was found to be connected with the HMBC of Cβ and Cγ, and Hγ (H4.18), Hγ (H3.78) was found to be correlated with the HMBC of Cα and Cβ, showing the existence of β-1 bonds in lignin[32]. The average molecular weight of lignin was 3990 by Fourier transform ion cyclotron resonance mass spectrometry[33]. Based on the above information, the possible molecular structure of lignin is shown in Supplementary Fig. 5. ChemDraw analysis results show that the chemical formula of lignin is $C_{205}H_{234}O_{81}$, and its element content (C: 61.65, H: 5.91, O: 32.45) is consistent with the actual test results (Supplementary Fig. 6 and Supplementary Table 3).

The types of lignin pyrolysis products depend on the temperature of the reactor and the residence time in the reactor[34]. The DTG curve shows that the significant thermal weight loss of lignin occurs between 200 and 500 °C, and the maximum weight loss temperature is about 360 °C (Supplementary Fig. 7). When the temperature rises to 1400 °C, the residual coke rate of lignin is 40%. At 1400 °C, the accumulated gaseous small molecule products from lignin pyrolysis are mainly $H_2O$ (57%), $H_2$ (23.82%), CO (14.24%), $CH_4$ (2.14%), $CO_2$ (2.8%) (Fig. 2b).

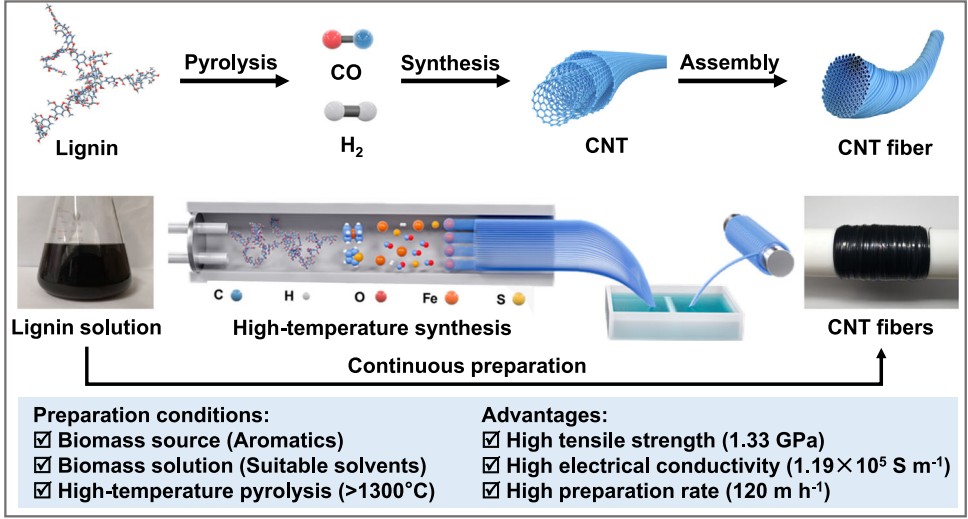

**Fig. 1 | Continuous production of CNT fibers.** Synthesis mechanism from lignin to CNT fibers and its respective processing schematic.

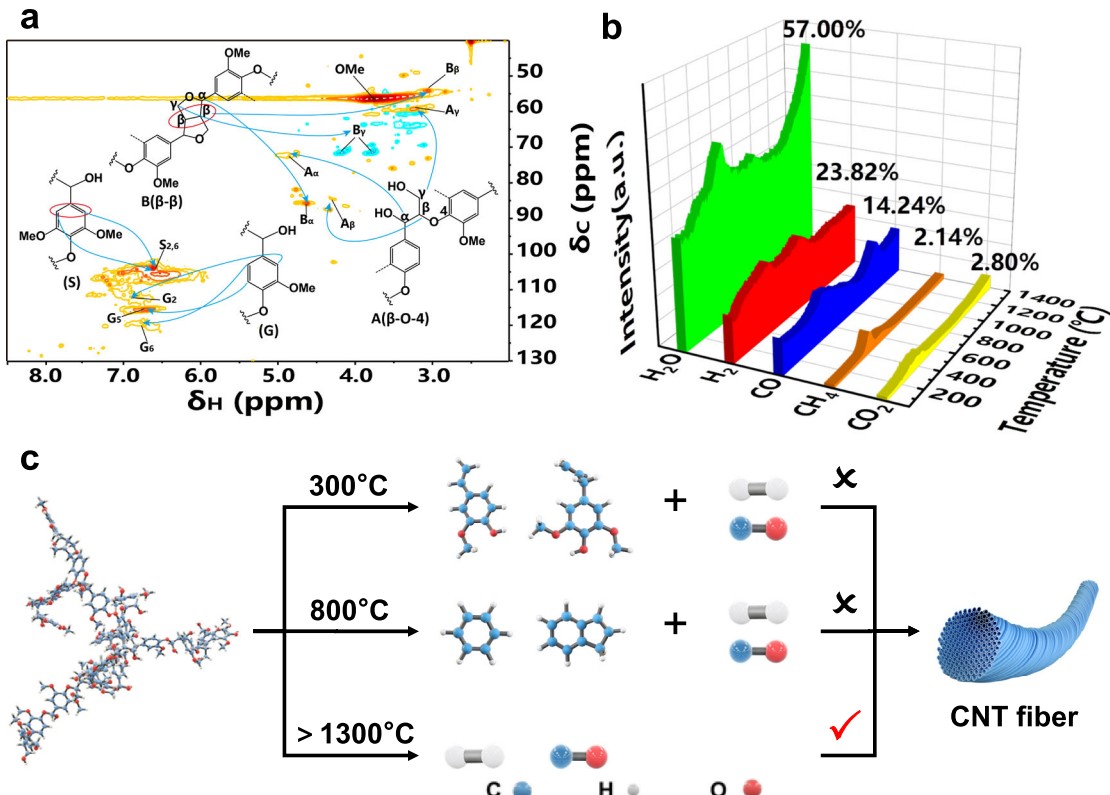

**Fig. 2 | Pyrolysis of lignin and synthesis conditions of CNT fibers. a** 2D NMR HSQC spectra (in DMSO-d6) of lignin. **b** Content of main products obtained from lignin pyrolysis (50–1400 °C) by TG-MS analysis. **c** Schematic showing the conditions under which CNT fibers can be obtained.

CO mainly comes from the cleavage of ether bonds in the side chains and between the aromatic rings in lignin, as well as the secondary decomposition of some volatiles. $CO_2$ mainly comes from the cleavage and reformation of reactive functional groups (such as carbonyl and carboxyl groups) in the side chains. $CH_4$ is derived from the side-chain cleavage and demethylation of methoxy groups on the benzene rings[35]. $H_2O$ is mainly produced by the hydroxyl groups on the aliphatic side chains of lignin[36]. The formation of $H_2$ can be attributed to the rearrangement of broken bonds in the aromatic rings[37]. CO is an effective carbon source for the synthesis of CNTs[38], and its content is significantly higher than that of $CH_4$ and $CO_2$ due to the wide range of sources.

Pyrolysis gas chromatography-mass spectrometry (PY-GCMS) results show the structural changes of MAHs at different temperatures from 200 to 800 °C (Supplementary Fig. 8). The primary pyrolysis reactions of lignin are retro-ene and Maccoll reactions at 500–600 °C and C−O homolysis reaction above 1000 °C[34]. At 800 °C, small molecular gases and aromatic ring compounds coexist in the lignin pyrolysis products (Supplementary Fig. 9 and Supplementary Tables 4 and 5). Small molecular gases may be produced by the removal of side groups of aromatic hydrocarbons (Supplementary Fig. 10)[39,40].

For the pyrolysis of lignin at higher temperatures, FTIR was used to analyze the high-temperature carbon in the inert atmosphere (Supplementary Figs. 11 and 12). The results show that there is no organic matter in the carbides obtained from lignin pyrolysis at 1400 °C. Some studies have shown that polycyclic aromatic hydrocarbons (PAHs) can be obtained by the reversible pyrolysis and polycondensation of lignin at high temperatures[41,42]. Our study shows that PAHs, such as naphthalene, can be obtained from lignin pyrolysis at 800 °C, while only MAHs can be obtained below 700 °C. The formation of PAHs easily leads to carbon deposition, which is an important cause of catalyst deactivation. At high temperatures above 1300 °C, lignin can be completely decomposed into small molecular gases, thus

avoiding carbon deposition and catalyst deactivation (Fig. 2c). In addition, it has been reported that the quality, growth rate, and length of CNTs can be improved in the presence of oxygen-containing additives, such as $H_2O$[43]. Therefore, the water produced by lignin pyrolysis in our system is also beneficial for the synthesis of CNTs.

When the reaction temperature is 400–1300 °C, carbon source CO for the synthesis of CNTs can be generated, but CNTs cannot be obtained under these conditions (Fig. 2c). On the one hand, the ratio of carbon source to hydrogen is very important for CNT synthesis. When the temperature is below 1300 °C, lignin cannot be completely decomposed into small molecular gases and there are a large number of aromatic compounds. Therefore, there is not enough CO in the pyrolysis products to generate CNTs. On the other hand, when the temperature is higher than 900 °C, lignin will undergo carbon reforming, which requires high activation energy of 178 kJ mol$^{-1}$. The C=C bonds do not begin to decompose until above 1000 °C and they can only be completely broken at about 1400 °C[44]. Therefore, our experimental results show that only when the synthesis temperature of CNTs is raised to above 1300 °C can the high pyrolysis activation energy of lignin be satisfied, so as to realize the complete pyrolysis of lignin and the continuous preparation of CNTs.

## Synthesis and structures of CNTs

Figure 3 shows the morphology and structure of the synthesized CNTs from lignin. Under the action of iron catalysts, the small molecule gases produced by lignin pyrolysis continuously generate CNT aggregates (Fig. 3a, b). The CNTs have an average outside diameter (OD) of 34 nm (Fig. 3c, d). TEM images show that the prepared CNTs are multi-walled (Fig. 3e, f). The CNTs have an $I_G/I_D$ value of 3.84 (Fig. 3g), which is higher than or similar to that of multi-walled CNTs (MWNTs) prepared from fine chemicals and other biomass (Supplementary Table 1). The $I_G/I_D$ values of MWNTs are generally lower than those of single-walled CNTs (SWNTs) and double-walled CNTs

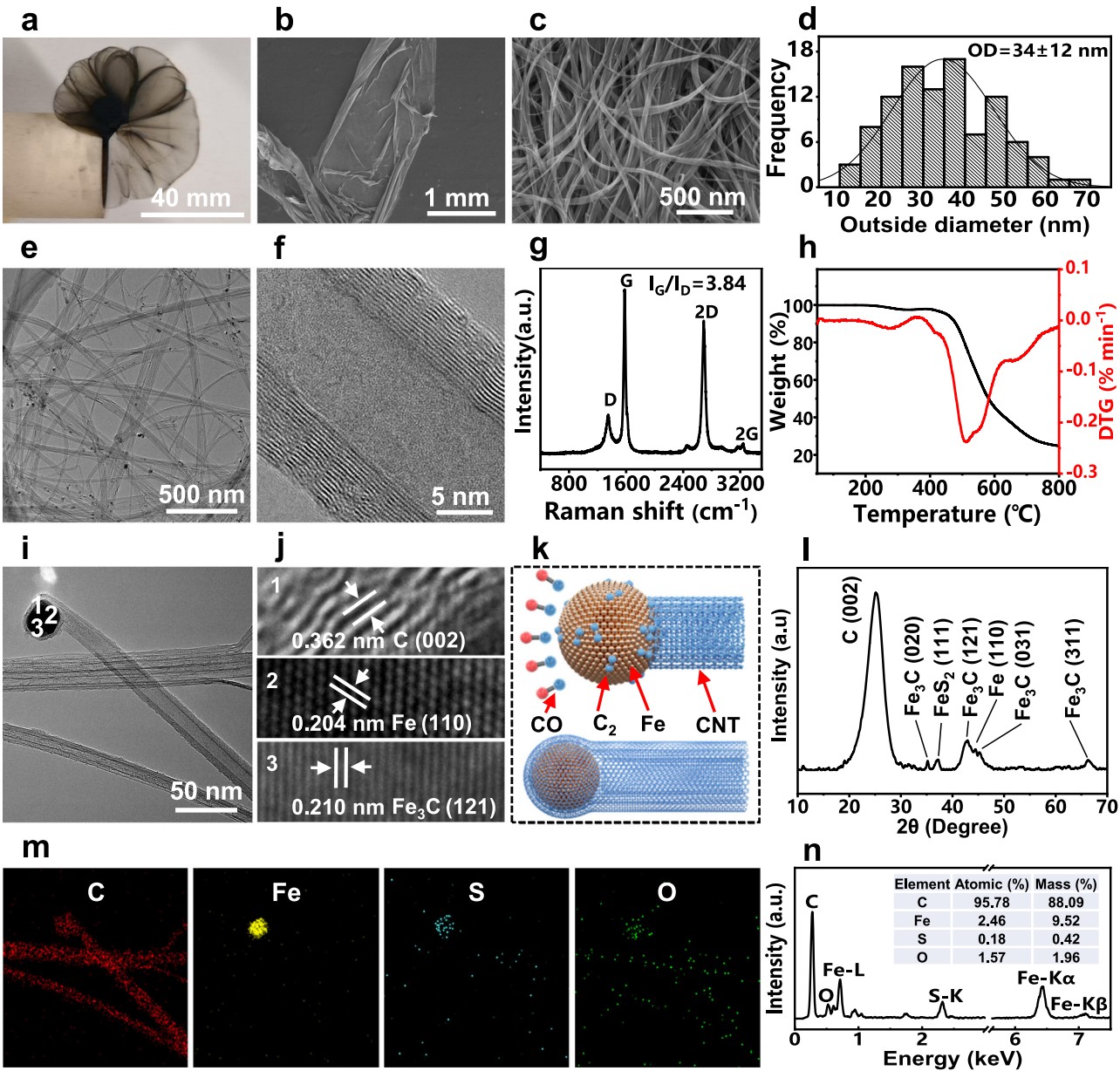

**Fig. 3 | Synthesis and structures of the lignin-derived CNTs. a** Digital image and **b** SEM image of a CNT sock. **c** SEM image and **d** diameter distribution of the CNTs. **e, f** TEM images of the CNTs. **g** Raman spectrum and **h** thermogravimetric analysis (TGA) of the CNTs. **i** TEM image of the CNTs and catalyst particles. **j** Lattices of the CNTs and iron catalysts. **k** Schematic showing the growth mechanism of the CNTs on iron catalysts. **l** XRD pattern of the CNTs. **m** Elemental mapping images of the CNTs. **n** TEM-EDS image of the CNTs and the element contents.

(DWNTs) due to the edge unsaturated carbon atoms, asymmetric carbon atoms, and sidewall structural defects in the MWNTs[45,46]. These functional groups and defects will be removed at high temperatures (300–400 °C), thus exhibiting mass loss of approximately 2.3% in the TGA result (Fig. 3h). TGA result also shows that the mass fraction of the CNTs in the aggregates is 82.7%, which is similar to the purity of the CNTs prepared by the same method[47]. Note that Fe in the sample was converted to $Fe_2O_3$ when heated at a high temperature in air, so the removal of oxygen in $Fe_2O_3$ is required to calculate the impurity content[48]. In addition, based on the carbon content (61.9%) and feeding rate (4.8 mg min$^{-1}$) of lignin as well as the preparation rate (1.46 mg min$^{-1}$) and purity (82.7%) of the CNT aggregates, the yield of the CNTs is about 40.6%.

The formation mechanism of the CNTs is as follows: At a temperature greater than 1300 °C, CO obtained from lignin pyrolysis first adsorbs on the surface of iron catalyst particles and forms C-C dimers.

Then the dimers leave the surface of the catalysts and link to each other to form short chains. Finally, these short chains are connected to each other to form sp2 bonded graphene sheets (Fig. 3i–k). The growth model of our CNTs is the so-called "cap" type, that is, all layers of the CNTs are deposited on the surface of the catalysts at the same time[49,50].

In the process of CNT synthesis, the sulfur atoms from thiophene pyrolysis are adsorbed on the surface of the iron catalysts and form sulfur-rich microregions. Because these microregions have lower surface energy than iron particles, the addition of sulfur will promote the deposition of carbon atoms on the surface of iron particles[51,52]. Due to the presence of iron catalysts and sulfur, Fe, $Fe_3C$, and $FeS_2$ crystals can be observed in the CNT aggregates (Fig. 3l). In addition, the presence of iron and sulfur atoms was also demonstrated through elemental mapping images and TEM-EDS analysis (Fig. 3m, n).

We removed thiophene from the CNT synthesis solution and performed CNT synthesis without sulfur. Under this condition, soot-

like substances are formed in the reactor (Supplementary Fig. 13a). Due to the absence of sulfur, the surface energy of the iron catalysts is high and the carbon atoms have no depositing sites on the catalysts, so crystalline carbon is formed and wrapped on the whole surface of the catalysts. The synthesized products include a large number of carbon nanospheres and a small number of CNTs (Supplementary Figs. 13b, c). The formation of crystalline carbon can be verified by Raman spectroscopy and the prepared carbon nanospheres have an $I_G/I_D$ of 3.81 (Supplementary Fig. 13d). The TGA results show that the mass fraction of the carbon nanospheres and CNTs is about 37.2% (Supplementary Fig. 13e). The peak at 450 °C on the TG curve is caused by the formation of $Fe_2O_3$ from the reaction of iron and oxygen.

When excessive thiophene is added to the CNT synthetic solution, too much sulfur will cause excessive deposition of carbon on the surface of the catalysts, resulting in the formation of curled CNTs (Supplementary Figs. 14a–c). Under this condition, CNT flocs are formed in the reactor. It's difficult for these resulting CNTs to assemble into thick hollow cylinders and further form CNT films. Raman results show that the obtained CNTs are amorphous, and they have an $I_G/I_D$ of 1.20 (Supplementary Fig. 14d). In addition, TGA shows that most of the products are CNTs, and the mass fraction of the CNTs is about 87.5% (Supplementary Fig. 14e).

In order to study the effect of solvent dispersion on the synthesis of CNTs, the mixture of lignin and catalysts were directly placed in a tubular furnace at 1400 °C for 30 min. Only black charcoal was obtained under these conditions, CNTs and carbon nanospheres were not hardly formed. The obtained black charcoal is partially crystallized and have an $I_G/I_D$ of 0.68 (Supplementary Figs. 15 and 16). Previous studies have also shown that only a small number of MWNTs can be produced using this approach, mostly in the form of amorphous carbon[28].

The effect of lignin concentration on the synthesis of CNTs was investigated. With the decrease of lignin concentration, the amount of product produced from the tubular furnace decreased gradually (Supplementary Fig. 17). When the lignin concentration decreased to 0.8 mg mL$^{-1}$, CNT fibers cannot be prepared continuously (Supplementary Fig. 17j). When the lignin concentration is lower than 0.4 mg mL$^{-1}$, only a very small amount of product can be formed (Supplementary Fig. 17k). For the solution without lignin, the product cannot be observed at the outlet and inside of the tubular furnace (Supplementary Fig. 17i). These results indicated that the CNTs were synthesized from lignin, and pure methanol cannot be used as a carbon source to prepare CNTs, which is in line with some reported work[53,54].

We also investigated the effect of lignin concentration on the morphology of CNTs. The results show that SWNTs and DWNTs can be obtained when the lignin concentration is lower than 0.8 mg mL$^{-1}$ (Supplementary Fig. 18). The acquisition of SWNTs and DWNTs can be attributed to the reduced amount of carbon deposited on the surface of the Fe catalyst. The existence of radial breathing mode (RBM) stretching vibration peak (100–300 cm$^{-1}$) in Raman spectrum also proved the synthesis of SWNTs (Supplementary Figs. 18d, e). When the lignin concentration is higher than 5.5 mg mL$^{-1}$, the excessive lignin will cause too many wall layers of CNTs, and a large number of carbon nanorods and amorphous carbon spheres also can be formed (Supplementary Figs. 18j–l).

In the process of CNT synthesis, the way lignin solution is injected into the tubular furnace depends on the solution concentration. When the concentration of lignin solutions is less than 0.5 mg mL$^{-1}$, lignin can be completely dissolved in methanol. No lignin precipitate can be found in these solutions after standing at room temperature for 12 h (Supplementary Fig. 19a). These low-concentration lignin solutions can be injected directly into the tubular furnace for CNT synthesis. When the solution concentration increases to more than 0.5 mg mL$^{-1}$, lignin cannot be completely dissolved. After standing for 12 h at room

temperature, lignin precipitates from the solutions obtained by magnetic stirring (Supplementary Fig. 19b). The amount of lignin precipitation depends on the solution concentration, and the higher the concentration, the more lignin is precipitated. For these high-concentration lignin solutions, they should be continuously oscillated to keep them in a uniform dispersion state during the process of CNT synthesis (Supplementary Fig. 19c).

The effect of injection rate (1–10 mL min$^{-1}$) of lignin solution with a concentration of 2.5 mg mL$^{-1}$ on the CNT fiber preparation was further studied. When the injection rate is lower than 1.5 mL min$^{-1}$, it's difficult to observe solid formation in the tubular furnace. When the injection rate is in the range of 1.5–2.5 mL min$^{-1}$, a small amount of CNT aerogels can be synthesized, but CNT fibers cannot be continuously prepared. The optimal injection rate range is 2.5–4.5 mL min$^{-1}$, in which CNT fibers can be continuously prepared. When the injection rate is higher than 4.5 mL min$^{-1}$, it is easy for the lignin solution to form aggregation and spray flame, which makes CNT fiber preparation unstable.

## Structures and properties of CNT fibers

The nascent CNT fibers (NCFs) were obtained from the compaction and winding of the synthetic lignin-derived CNTs (Supplementary Fig. 20a). These CNT fibers have a loose structure and low degree of orientation (Supplementary Fig. 20b). In order to improve the mechanical properties of the CNT fibers, they were treated by two methods, including twisting and rolling (Fig. 4a, b). The twisted CNT fibers (TCFs) with a diameter of about 38 μm achieved a high degree of densification, and the spiral patterns can be clearly observed (Fig. 4c). As can be seen from the cross-section of the TCFs, the inside of the fibers is not dense enough (Supplementary Fig. 20c, d), which results in a low density of 0.64 g cm$^{-3}$ (Supplementary Fig. 21a, b). In addition, excessive twisting cannot improve the mechanical strength of the CNT fibers, but cause fiber fracture[55].

Compared with twisting, CNT fibers prepared by rolling (rolled CNT fibers, RCFs) have a denser structure (Fig. 4d and Supplementary Fig. 20e) and improved fiber orientation (Supplementary Fig. 20f) due to the greater stress applied to the CNT fibers. The density of the RCFs is as high as 1.49 g cm$^{-3}$ (Supplementary Fig. 21c–e). The orientation of the CNT fibers was measured by the azimuth scanning of wide-angle X-ray diffraction (WAXD) (Supplementary Fig. 22). The full width at half maximum (FWHM) of the TCFs and RCFs are 71.58° and 60.08°, respectively, much smaller than that of the NCFs (Supplementary Fig. 23). Note that the FWHM of the NCFs is 91.32°. The CNT fibers treated by densification have a higher orientation degree along the axial direction than the NCFs, and the RCFs exhibit better orientation than the TCFs. In addition, the $I_{G\parallel}/I_{G\perp}$ values of the TCFs and RCFs obtained from polarized Raman spectra are 1.35 and 2.45, respectively, indicating that RCFs have better orientation than TCFs (Fig. 4e, f).

CNT fibers obtained by different densification methods have different mechanical properties. The tensile strength of TCFs and RCFs are 0.27 ± 0.02 GPa and 1.33 ± 0.09 GPa, respectively, and their elastic moduli are 10.46 ± 1.24 GPa and 37.45 ± 7.47 GPa, respectively (Fig. 4g). RCFs have a denser structure and a more oriented structure compared to TCFs, which results in higher friction and more difficult slippage between CNTs in the fibers, thus achieving significantly better mechanical properties. The elongation at break of TCFs and RCFs are 6.12 ± 0.43% and 5.62 ± 0.18%, respectively. Although TCFs and RCFs have a similar elongation at break, RCFs exhibit significantly higher fracture work due to their significantly higher mechanical strength. Note that the fracture work of the TCFs and RCFs are 10.89 ± 1.13 MJ m$^{-3}$ and 47.54 ± 3.85 MJ m$^{-3}$, respectively (Fig. 4h).

In addition to excellent mechanical properties, the CNT films with a density of 0.82 g cm$^{-3}$ exhibit high thermal conductivity of 33.21 ± 0.76 W m$^{-1}$ K$^{-1}$ (Supplementary Fig. 24). Compared to biomass-derived carbon materials (0.06–24 W m$^{-1}$ K$^{-1}$), our CNT films possess higher thermal conductivity, comparable to that of the CNT

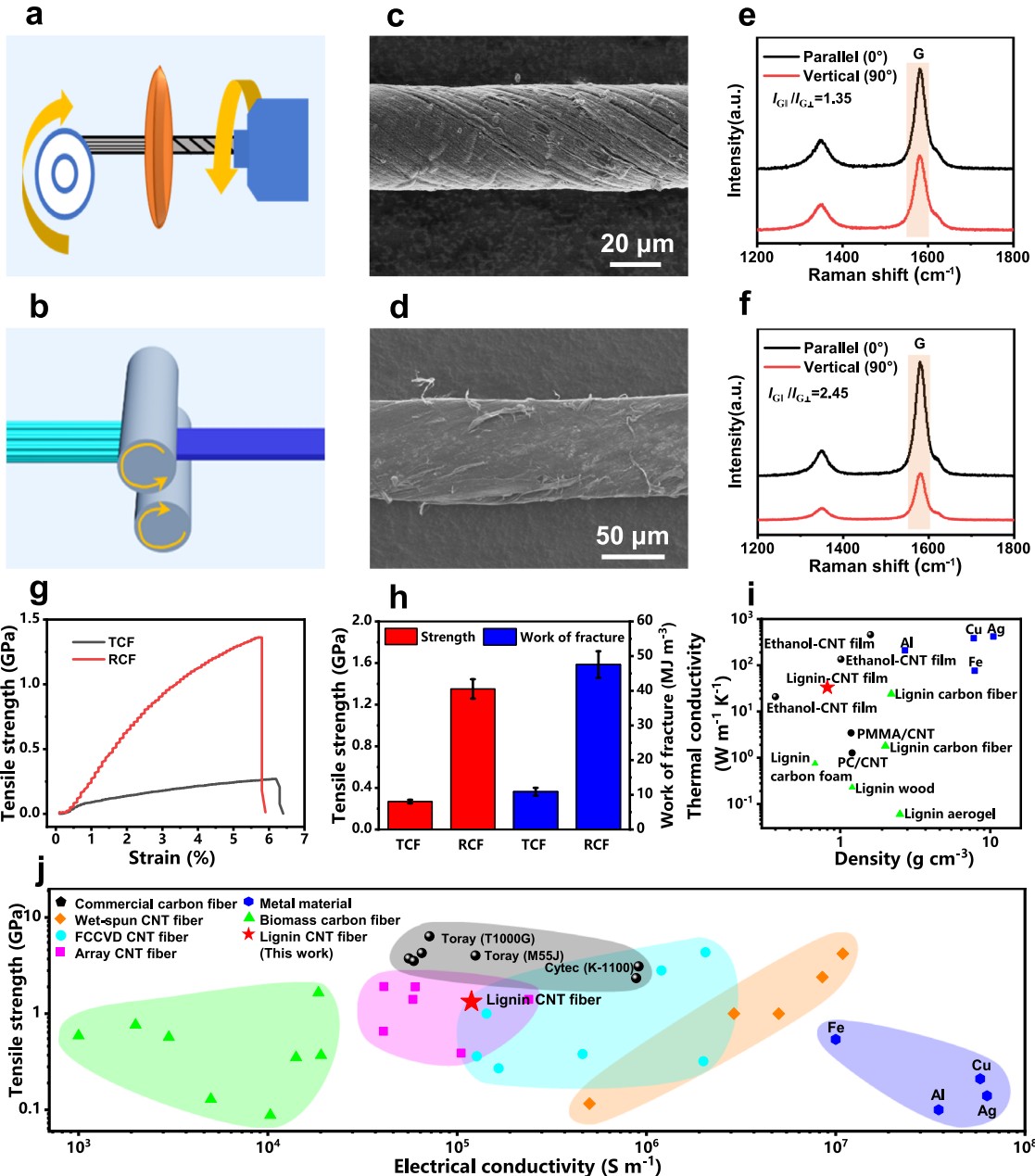

**Fig. 4 | Preparation and properties of lignin-derived CNT fibers.** Preparation diagrams of **a** TCFs and **b** RCFs. SEM images of **c** TCFs and **d** RCFs. Polarized Raman spectra of **e** TCFs and **f** RCFs. **g** Tensile stress-strain curves of the TCFs and RCFs. **h** Comparison of tensile strength and fracture work for the TCFs and RCFs. Error bars represent s.d. ($n = 5$). **i** Comparison of the thermal conductivity and density between our CNT films and other materials[48,60–67]. **j** Comparison of the tensile strength and electrical conductivity of our CNT fibers with commercial carbon fibers[68], wet-spun CNT fibers[18,69–71], CNT fibers from FCCVD[25,56,72–74], CNT fibers from array spinning[75–79], common metal materials[67], and biomass-derived carbon fibers[80–86].

films with similar characteristics prepared by the similar method (20.91–458.58 W m$^{-1}$ K$^{-1}$) as well as some common metals (30–500 W m$^{-1}$ K$^{-1}$) (Fig. 4i and Supplementary Table 6). Considering that the CNT films have a significantly lower density (0.82 g m$^{-3}$) than common metals (2.7–10.49 g cm$^{-3}$), they can be used in some fields that require lightweight thermal conductive materials.

We also proved that our CNT fibers have high electrical conductivity, and the electrical conductivity of the CNT fibers with a density of 1.49 g cm$^{-3}$ is as high as $(1.19 \pm 0.09) \times 10^5$ S m$^{-1}$ (Supplementary Fig. 25), which is similar to that of the CNT fibers with similar structures prepared by the similar method[56]. The electrical conductivity of our CNT fibers is higher than that of almost all reported biomass-derived carbon fibers and array CNT fibers as well as most

commercial carbon fibers (Fig. 4j). It is worth noting that the electrical conductivity of our CNT fibers is lower than that of the wet-spun CNT fibers, which may be due to the higher purity and crystallinity of the CNTs used for wet-spinning as well as the higher density of the resultant CNT fibers (Fig. 4j and Supplementary Table 7).

Although the mechanical strength of the prepared CNT fibers is not yet comparable to that of commercial carbon fibers, it is higher than or similar to that of most reported biomass-derived carbon fibers, array CNT fibers, CNT fibers from FCCVD and wet-spun CNT fibers, as well as all common metal materials (Fig. 4j and Supplementary Table 7). It should be emphasized that the mechanical strength of our CNT fibers exceeds that of most CNT fibers prepared with fine chemicals (such as alkanes and aromatic hydrocarbons) as carbon sources.

Taken together, our lignin-derived CNT fibers show the integration of high tensile strength, thermal conductivity, and electrical conductivity, as well as a continuous preparation process.

## Discussion

There are few heat treatment steps to prepare CNT fibers by FCCVD method with lignin as carbon source, and the energy consumption is mainly concentrated in the step of lignin pyrolysis. The energy consumption of lignin-based CNT fibers prepared by our method was estimated to be about 0.12 MJ m$^{-1}$ (Supplementary Fig. 26). However, the high-temperature heat treatment steps of lignin-based carbon fibers prepared by traditional methods include pre-oxidation, carbonization and graphitization, with energy consumption up to 0.22–0.67 MJ m$^{-1}$[57,58].

In terms of production efficiency, it only takes 2 min to get CNT fibers from the feeding of lignin solution with our method, and the fiber preparation rate is up to 120 m h$^{-1}$. Compared with the preparation of CNT fibers using fine chemicals as raw materials, the preparation efficiency of our method is lower because it takes a certain amount of time for lignin to decompose into small molecules (Supplementary Table 8). However, the production of traditional lignin-based carbon fibers involves spinning and multistep heat treatment. It takes at least 90 min to get lignin-based carbon fibers, and the fiber preparation rate is only 20–35 m h$^{-1}$ (see ref. [57]). The high preparation efficiency and low energy consumption combined with low lignin pretreatment requirements make our method very promising for large-scale production of lignin-based CNT fibers.

In addition to lignin, our preparation method is also applicable to other aromatic biomass materials. Tea polyphenols with a molecular weight of 281 and tannic acid with a molecular weight of 1701 were also used as carbon sources and achieved continuous preparation of CNT fibers (Supplementary Figs. 27 and 28). These biomass materials were also dispersed in methanol-containing catalysts to synthesize CNT fibers at 1400 °C. The diameters of the CNTs prepared from tea polyphenols and tannic acid are 5–40 nm and 5–65 nm, respectively, and their $I_G/I_D$ values obtained by Raman spectra are 2.43 and 2.17, respectively. The wide applicability of our method to biomass materials will greatly promote the high-value utilization of various biomass materials.

Despite the advantages of our approach mentioned above, we have to admit that the mechanical strength of the CNT fibers prepared by FCCVD method using lignin as a carbon source is lower compared with traditional carbon fibers. We explored two methods to prepare CNT fibers, including twisting and rolling in this work. The porosities of the CNT fibers obtained by the two methods are 69.5% and 29.0%, respectively, and the corresponding tensile strength are 0.27 GPa and 1.33 GPa, respectively. The decrease of porosity significantly improves the tensile strength of the CNT fibers due to the increased mechanical interlocking between CNTs. Although the method for improving the mechanical properties of CNT fibers by acid treatment and heat treatment have been reported[47,59], the additional processes inevitably increase the cost of fiber manufacturing and reduce productivity, and are not conducive to the continuous preparation of CNT fibers. Therefore, for the preparation of CNT fibers with high mechanical properties from biomass resources, it is necessary to further explore new methods of fiber densification.

Suitable catalyst concentration is very important for the continuous preparation of CNT fibers. In our work, we use ferrocene as the catalyst, and the concentration of ferrocene in the lignin solutions is 0.005 g mL$^{-1}$, lower than that used in many literatures for the preparation of CNTs by similar methods (Supplementary Table 9). In addition, the amount of catalyst is also very important to control the production cost of CNT fibers. In order to further control the production cost of CNT fibers, the following aspects can be considered: (1) optimize the feeding method of lignin solutions, and explore the use of spray solution supply method to improve the efficiency of catalyst; (2) adjust the flow rate of carrier gas to prolong the residence time of catalyst in the reaction zone, thus improving the catalytic efficiency and reducing the amount of catalyst; (3) reduce the amount of catalyst by optimizing the proportion of each component in the lignin solutions.

In conclusion, we proved that biomass resources can be used as raw materials for CNT fibers, and realize continuous preparation of CNT fibers using FCCVD method. As a proof of concept, lignin was used as the carbon source and high-performance CNT fibers were developed. In order to achieve continuous preparation of CNT fibers, it's necessary to dissolve lignin in appropriate solvents, and the synthesis reaction needs to be completed at a high temperature of more than 1300 °C under the action of catalysts. The lignin-derived CNT fibers unprecedently integrated a high tensile strength of 1.33 GPa, a high thermal conductivity of 33.21 W m$^{-1}$ K$^{-1}$, and a high electrical conductivity of 1.19 × 10$^5$ S m$^{-1}$, exceeding that of most currently reported biomass-derived carbon materials. More importantly, the continuous preparation of lignin-derived CNT fibers combined with the production rate of 120 m h$^{-1}$ makes their large-scale preparation possible, which will greatly promote the application of lignin materials in high-end fields.

## Methods

### Materials and chemicals

Lignin (hardwood kraft lignin) was purchased from Suzano Papel e Cellulose S.A. in Brazil, and composition analysis of lignin is shown in Supplementary Table 10. Anhydrous methanol (AR, Sinopharm) was purchased from Sinopharm Chemical Reagent Co., LTD. Ferrocene (AR, Rhawn) and thiophene (AR, Innochem) were purchased from Wendong (Shanghai) Chemical Co., LTD. in China.

### Synthesis of CNT fibers

In all, 0.1–2.5 g lignin was dissolved in 250 mL methanol and stirred magnetically for 12 h. Then 0.5 g ferrocene and 0.5 g thiophene were added into the 100 mL lignin solution, and dissolved ultrasonically for 20 min to obtain uniform mixed solution. The lignin-mixed solution was pumped into a tubular furnace using a metering pump at a rate of 1–10 mL min$^{-1}$. High-purity argon gas was used as a carrier gas to carry the reactants forward continuously, and the reactants reacted in the process of movement. The airflow velocity of argon was set to 100 mL min$^{-1}$. The temperature in the tubular furnace was set at 400–1400 °C. Unless otherwise stated, CNTs were synthesized at 1400 °C. The CNT socks generated from the tubular furnace were immersed in water and densified to obtain nascent CNT fibers (NCFs). The same process was also used to prepare CNT fibers from tea polyphenol and tannic acid.

### Characterization

The hydrogen spectra ($^1$H NMR), carbon spectra ($^{13}$C NMR), hydrogen correlation spectroscopy ($^1$H–$^1$H COSY), two-dimensional heteronuclear single quantum coherence (2D-HSQC) spectra, and heteronuclear multiple bond coherence (HMBC) spectra were obtained by 600 MHz nuclear magnetic resonance (NMR) spectrometer to analyze the molecular structure of lignin. The pyrolysis products of lignin were characterized by PY-GCMS (QP 2010, Shimadzu, Japan). The decomposition process of lignin was analyzed by thermogravimetric-mass spectrometry (TG-MS, Thermo Plus EV2, Rigaku, Japan). The morphologies and microstructures of the samples were characterized by scanning electron microscope (SEM, SU8010, Hitachi, Japan) and transmission electron microscope (TEM, FEI Talos F200S, Hillsboro, USA). The CNTs purity was determined by a thermogravimetric analyzer (TGA8000, Perkin Elmer, USA) at a temperature range of 50–800 °C and a heating rate of 10 °C min$^{-1}$ in air. Polarized Raman spectroscopy (RM2000, Renishaw, UK) was used to analyze the graphitization degree and

orientation of the CNT fibers. The crystal structures of CNTs were characterized using X-ray diffractometry (XRD, Ultima IV, Rigaku, Japan) in the scanning range of 10–70° at a scanning speed of 2° min⁻¹. The orientation of the CNT fibers was measured by the azimuth scanning of wide-angle X-ray diffraction (WAXD, D/max 2550 VB/PC, Rigaku, Japan). The thermal conductivity of the CNT films was measured by a laser thermal conductivity analyzer (LFA467, Netzsch, Germany). The electrical conductivity of the CNT fibers was measured using a four-probe tester (RTS-9, Guangzhou Four Probe Technology Co., Ltd., China). The electrical conductivity ($\sigma$, S m⁻¹) was calculated by the following equation: $\sigma = L/(RS)$, where L is the distance between the four probes ($L = 1$ mm), $R$ is the resistance of the CNT fibers ($\Omega$), and $S$ is the cross-sectional area (m²). RCFs with a rectangular cross-section were used for the electrical conductivity determination, and ten samples were determined to obtain an average value. The mechanical properties of the CNT fibers were determined using a universal tensile testing machine (YG-004, Dahua Electronic, China), and the gauge length was set as 10 mm. The density of the CNT fibers is calculated based on their mass and volume. The porosity ($\varphi$) of the CNT fibers is calculated according to the following equation: $\varphi(\%) = \frac{\rho_{CNT} - \rho_{CNT\ fiber}}{\rho_{CNT}} \times 100\%$, where $\rho_{CNT}$ and $\rho_{CNT\ fiber}$ are the densities of pure CNT ($\rho_{CNT} = 2.1$ g cm⁻³) and CNT fibers, respectively. The mass per unit length is calculated based on the mass and length of the CNT fibers.

## Data availability

All the data supporting the findings of this study are available within the main text of this article and its Supplementary Information. Source data are provided with this paper.

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

## Acknowledgements

This study was supported by the Science and Technology Commission of Shanghai Municipality (20JC1414900), the Joint Funds of the National Natural Science Foundation of China (U20A20257), Program of Shanghai Academic/Technology Research Leader (20XD1433700), the Innovation Program of Shanghai Municipal Education Commission (2017-01-07-00-03-E00055), and the Fundamental Research Funds for the Central Universities (2232022D-04).

## Author contributions

M.Z., H.X., and C.J. conceived the idea and supervised the research. F.L., Q.W., G.Z., J.Z., and Q.W. contributed to the material preparation and characterization. F.L., C.J., H.X., and L.Z. contributed to the writing of the manuscript. All authors reviewed and commented on the manuscript.

## Competing interests

The authors declare no competing interests.
