## [Peer Review File · Nature Communications]

Continuously processing waste lignin into high-value carbon nanotube fibersREVIEWER COMMENTS

Reviewer #1 (Remarks to the Author):

The authors fabricated continuous CNT fibers using renewable biomass material as carbon source. The resulting CNT fibers possessed a tensile strength of 1.35 GPa and an electrical conductivity of $6.28 \times 10^5 \text{ S m}^{-1}$, which are much better than those of most biomass-derived carbon materials. Importantly, the production rate of fabrication process was impressively high (120 m h^{-1}). Moreover, their method was applicable to other biomass materials, suggesting the high value application of biomass in a wide range of fields. The work is interesting and can be published in Nature Communications if the following issues can be addressed:

1. This research work applied post-treatments to enhance properties of the CNT fibers and compared them with fibers fabricated from different methods. Therefore, review on fabrication methods for CNT fibers and post-treatment methods should be presented in the introduction: Three main methods to fabricate CNT fibers should be briefly mentioned: (1) wet-spinning (DOI: 10.1126/science.1094982), (2) array spinning (DOI: 10.1126/science.1104276), and (3) floating catalyst/aerogel spinning (DOI: 10.1126/science.1094982). Different post-treatment methods also need to be presented:
 - Solvent/mechanical densification (<https://doi.org/10.1016/B978-0-08-102722-6.00006-7>)
 - Chemical doping (<https://doi.org/10.1016/j.carbon.2021.08.024>)
 - Metal coating (<https://doi.org/10.1016/j.jmst.2019.08.057>)
 - Acid treatment (<https://doi.org/10.1016/j.apsusc.2013.11.162>)
 - Purification (<https://doi.org/10.1021/acsami.7b09287>)
2. From TGA results in Figure 3h, the authors mentioned that the CNT fibers were highly pure with a CNT mass fraction of 75.3%. However, an impurity of 24.7% is considered significantly high when some other studies reported the array-spun, aerogel-spun, and wet-spun CNT fibers with impurity below 5% (<https://doi.org/10.1038/srep03903>). The authors should clarify this issue.
3. The CNT diameter is quite large with many impurities. This could lower the performance of the CNT fibers. How the process parameters can be tailored to control the morphologies of the CNTs (for example, SWNT or DWNT with much lower impurity)?
4. Scale bar for Figure 3a is required.
5. The cross-sections in Figure S17c and d were obtained from distorted surfaces/areas and, therefore, should not be used to determine the density of the fiber structure.
6. The authors mentioned that the process had low energy consumption. How was the energy consumption estimated in this study? This should consider the post-treatment processes.
7. How was the porosity of the twisted and rolled fibers determined?

Reviewer #2 (Remarks to the Author):

Lignin is a bulk by-product of paper industry, whereas has not been available used so far. In this work, high-performance CNT fibers were prepared by using industrial lignin as the carbon source under a highly continuous production speed. This paper was well written and the data was sufficient. This work would provide a new idea for the high-value utilization of industrial lignin or other biomass-based materials. However, some parts within the manuscript still needed to be improved. I think that it could be acceptable in Nature Communications after addressing the following comments:

1. The authors have provided the structural characterizations of the raw lignin, as well as the effects of functional groups and chemical bonds on the pyrolysis of lignin. However, I still wonder whether there are C-C bonds between the aromatic rings of the raw lignin?

2. This work investigated the utilization of lignin as a carbon source for the synthesis of CNTs. The carbon atoms existed in the side chain, benzene ring and methoxy groups of lignin macromolecules. So what structures in lignin does H₂O, H₂, CO, CH₄ and CO₂ derived from? Why is the yield of CO greater than that of CO₂? The author should give a specific explanation in the manuscript.

3. How do the authors calculate the purity of the generated CNTs?

4. The authors have mentioned that the rolling method can make fibers denser than the twisting method. The author should explain the corresponding mechanism in the text.

5. Various amounts of lignin were applied in the synthesis of CNTs. So what is the effect of these variables on the resulting CNTs? Furthermore, how does the injection rate (1-10 mL min⁻¹) affect on products?

6. The authors used methanol as a solvent, whether the lignin can be completely dissolved in methanol? The authors should provide the data in the manuscript.

7. In the "test performance" section, what is the length-diameter ratio of CNTs fibers used for the mechanical strength test? What is the effect of different length-diameter ratios on mechanical strength?

8. Other important physical parameters such as density, mass per unit length, elastic modulus and elongation at break should be given in the manuscript.

9. In the synthesis process, the author used a large number of catalysts based on the quality of raw lignin. How to control production cost? How are these catalysts disposed after the reaction?

10. The author only compared the literatures of the biomass-derived carbon fiber materials. Comparisons with CNTs fibers generated by classical FCCVD methods should also be considered to support the superiority of this work.

Reviewer #3 (Remarks to the Author):

In this manuscript by Liu and co-workers, the authors investigated the possibility of synthesis of carbon nanotubes from kraft lignin solution in methanol. Based on this concept, a direct spinning variant of CVD was utilized to manufacture fibers in a continuous manner. The results are interesting, but some issues should be addressed first before the submission can be reconsidered for publication in Nature Communications. Please find suggestions below:

1) "The carbon sources used in this method are mainly from petroleum fine chemicals, such as methane, ethylene, ethanol and xylene" – aromatic solvents such as toluene should also be mentioned due to their widespread use

2) "After post-treatment, the lignin-based CNT fibers were endowed with a tensile strength of 1.35 GPa and an electrical conductivity of 6.28×10^5 S m⁻¹. In addition, the continuous production of CNTs fibers from lignin with a 120 m h⁻¹ production rate was achieved." - these values should be compared with the whole state of the art (not just narrowed down to CNTs synthesized from natural resources but also from synthetic precursors such as alkanes and aromatic hydrocarbons). Such a summary would be useful to evaluate how good the reported values really are

3) "TGA result shows that the mass fraction of the CNTs in the aggregates is 75.3% (Figure 3h), which indicates that the lignin-based CNT fibers have high purity" – judging by the provided thermogram, the nanotubes are of poor crystallinity. Yet, the authors report extremely high electrical conductivity and thermal conductivity. Please comment on this issue.

4) Regarding the electrical conductivity of CNT fibers, a primary source of error, which may greatly affect the result, is the cross-section area. Because the authors report very high electrical conductivity values (on the order of thousands of S/cm), more information should be provided on how these values were obtained (especially how the diameter was established). Currently, the

following description is not very informative "The determination of electrical conductivity was performed on a Digit Graphical Touchscreen Digital Multimeter (DMM6500 6½)". Was it a two- or four-probe approach?

5) Whenever possible, error analysis should be conducted. The absence of error bars casts doubt about the statistical significance of the reported data.

6) Minor comment, in Table S1, it is recommended to change "Layer number" to "CNT type". "MWNT" and "SWNT" are not numerical values.

Point-by-point Response Letter

We would like to thank the three reviewers for their constructive comments. We have carried out additional experiments and discussion to address the reviewers' comments point-by-point. Please find our detailed responses in the response letter.

Reviewer #1

The authors fabricated continuous CNT fibers using renewable biomass material as carbon source. The resulting CNT fibers possessed a tensile strength of 1.35 GPa and an electrical conductivity of $6.28 \times 10^5 \text{ S m}^{-1}$, which are much better than those of most biomass-derived carbon materials. Importantly, the production rate of fabrication process was impressively high (120 m h^{-1}). Moreover, their method was applicable to other biomass materials, suggesting the high value application of biomass in a wide range of fields. The work is interesting and can be published in Nature Communications if the following issues can be addressed.

Response: We thank the reviewer for the positive comments.

Comment 1: *This research work applied post-treatments to enhance properties of the CNT fibers and compared them with fibers fabricated from different methods. Therefore, review on fabrication methods for CNT fibers and post-treatment methods should be presented in the introduction:*

Three main methods to fabricate CNT fibers should be briefly mentioned: (1) wet-spinning (DOI: 10.1126/science.1094982), (2) array spinning (DOI: 10.1126/science.1104276), and (3) floating catalyst/aerogel spinning (DOI: 10.1126/science.1094982). Different post-treatment methods also need to be presented:

- Solvent/mechanical densification (<https://doi.org/10.1016/B978-0-08-102722-6.00006-7>)
- Chemical doping (<https://doi.org/10.1016/j.carbon.2021.08.024>)
- Metal coating (<https://doi.org/10.1016/j.jmst.2019.08.057>)
- Acid treatment (<https://doi.org/10.1016/j.apsusc.2013.11.162>)
- Purification (<https://doi.org/10.1021/acsami.7b09287>)

Response: We thank the reviewer for the comments.

Per the suggestion of the reviewer, we added the three main methods for fabricating CNT

fibers and their post-treatment methods in the revised manuscript.

Corresponding changes:

Page 4 in Manuscript:

The preparation methods of CNT fibers include wet spinning¹⁸, array spinning¹⁹, and floating catalyst chemical vapor deposition (FCCVD)²⁰. In addition, post-treatment is usually used to improve the mechanical, electrical and thermal properties of CNT fibers, including solvent/mechanical densification²¹, chemical doping²², metal coating²³, acid treatment²⁴, and purification²⁵.

Comment 2: *From TGA results in Figure 3h, the authors mentioned that the CNT fibers were highly pure with a CNT mass fraction of 75.3%. However, an impurity of 24.7% is considered significantly high when some other studies reported the array-spun, aerogel-spun, and wet-spun CNT fibers with impurity below 5% (<https://doi.org/10.1038/srep03903>). The authors should clarify this issue.*

Response: We thank the reviewer for the comments.

The original CNT fibers are mainly composed of C and Fe elements (Figs. 3l-n). In thermogravimetric analysis, C was completely removed from the sample, and only Fe was remained. In fact, Fe in the sample was converted to Fe₂O₃ when heated at high temperature in air¹, so the residue of the sample in Fig. 3h was Fe₂O₃. We recalculated the Fe content in the sample by removing oxygen. The actual Fe content (impurity content) in the sample is 17.3%. Therefore, the CNT mass fraction in the sample should be 82.7%.

Fig. 3 Synthesis and structures of the lignin-derived CNTs. (a) Digital image and (b) SEM image of a CNT sock. (c) SEM image and (d) diameter distribution of the CNTs. (e, f) TEM images of the CNTs. (g) Raman spectrum and (h) thermogravimetric analysis (TGA) of the CNTs. (i) TEM image of the CNTs and catalyst particles. (j) Lattices of the CNTs and iron catalysts. (k) Schematic showing the growth mechanism of the CNTs on iron catalysts. (l) XRD pattern of the CNTs. (m) Elemental mapping images of the CNTs. (n) TEM-EDS image of the CNTs and the element contents.

We fully investigated other common processes for preparing CNT fibers, including wet-spinning^{2,3}, array-spinning^{4,5}, and aerogel-spinning⁶. For the wet-spun CNT fibers^{2,3}, the first step is to dissolve CNTs in chlorosulfonic acid to remove impurities, so the CNT fibers obtained by this method has high purity. However, the preparation process is complicated, and chlorosulfonic acid has environmental pollution problem. For the array-spun CNT fibers^{4,5}, Fe is deposited on Si substrate to catalyze CNT synthesis, so the resulting CNT array has almost no Fe impurity. Although this method can obtain ultra-high purity CNT fibers, it is not only complicated, but also cannot achieve continuous preparation of CNT fibers. For the work mentioned by the reviewer (<https://doi.org/10.1038/srep03903>)⁶, although high purity CNTs (less than 5% impurity) can be obtained by aerogel-spinning, the impurity content of the CNTs

is more than 5% or even more than 15% in some conditions, similar to that obtained in our work. In addition, similar CNT purity was obtained by aerogel-spinning in other reported work⁷. It should be noted that it is possible to further improve the purity of CNTs by adjusting the synthesis process using lignin as carbon source.

In order to avoid misunderstanding, we deleted the statement that lignin-based CNT fibers have high purity in the revised manuscript.

Corresponding changes:

Page 9 in Manuscript:

TGA result shows that the mass fraction of the CNTs in the aggregates is 82.7% (Fig. 3h), which is similar to the purity of the CNTs prepared by the same method⁴⁵. Note that Fe in the sample was converted to Fe₂O₃ when heated at high temperature in air, so the removal of oxygen in Fe₂O₃ is required to calculate the impurity content⁴⁶. In addition, based on the carbon content (61.9%) and feeding rate (4.8 mg min⁻¹) of lignin as well as the preparation rate (1.46 mg min⁻¹) and purity (82.7%) of the CNT aggregates, the yield of the CNTs is about 40.6%.

References:

1. Zhan, H., Chen, Y. W., Shi, Q. Q., Zhang, Y., Mo, R. W. & Wang J. N. Highly aligned and densified carbon nanotube films with superior thermal conductivity and mechanical strength. *Carbon* **186**, 205-214 (2022).
2. Behabtu, N. et al. Strong, light, multifunctional fibers of carbon nanotubes with ultrahigh conductivity. *Science* **339**, 182-186 (2013).
3. Ericson, L. M. et al. Macroscopic, neat, single-walled carbon nanotube fibers. *Science* **305**, 1447-1450 (2004).
4. Zhang, M., Atkinson, K. R. & Baughman, R. H. Multifunctional carbon nanotube yarns by downsizing an ancient technology. *Science* **306**, 1358-1361 (2004).
5. Li, Q. W., et al. Sustained growth of ultralong carbon nanotube arrays for fiber spinning. *Adv. Mater.* **18**, 3160-3163 (2006).
6. Paukner, C. & Koziol, K. K. Ultra-pure single wall carbon nanotube fibres continuously spun without promoter. *Sci. Rep.* **4**, 1-7 (2014).
7. Lee, J. et al. Direct spinning and densification method for high-performance carbon nanotube fibers. *Nat. Commun.* **10**, 1-10 (2019).

Comment 3: *The CNT diameter is quite large with many impurities. This could lower the performance of the CNT fibers. How the process parameters can be tailored to control the morphologies of the CNTs (for example, SWNT or DWNT with much lower impurity)?*

Response: We thank the reviewer for the comments.

In our work, we studied the effects of synthesis temperature, thiophene content and solvent dispersion on the morphology of CNTs, as shown in Fig. 3, Supplementary Figs. 13, 14 and 15. In these studies, the obtained CNTs are multiwalled.

Fig. 3 Synthesis and structures of the lignin-derived CNTs. (a) Digital image and (b) SEM image of a CNT sock. (c) SEM image and (d) diameter distribution of the CNTs. (e, f) TEM images of the CNTs. (g) Raman spectrum and (h) thermogravimetric analysis (TGA) of the CNTs. (i) TEM image of the CNTs and catalyst particles. (j) Lattices of the CNTs and iron catalysts. (k) Schematic showing the growth mechanism of the CNTs on iron catalysts. (l) XRD pattern of the CNTs. (m) Elemental mapping images of the CNTs. (n) TEM-EDS image of the CNTs and the element contents.

Supplementary Figure 13. Carbon nanospheres synthesis without sulfur. (a) Digital image and (b, c) TEM images of the carbon nanospheres. (d) Raman spectrum and (e) TGA of the carbon nanospheres prepared without thiophene addition.

Supplementary Figure 14. CNT synthesis with excessive sulfur. (a) Digital image and (b, c) TEM images of the curled CNTs. (d) Raman spectrum and (e) TGA of the amorphous CNTs prepared with excessive thiophene addition.

Supplementary Figure 15. Raman spectrum of black charcoal from the pyrolysis of solid lignin at 1400°C for 30 min in N₂ atmosphere.

We also investigated the effect of lignin concentration on the morphology of CNTs. The results show that single-walled carbon nanotubes (SWNT) and double-walled carbon nanotubes (DWNT) can be obtained when the lignin concentration is lower than 0.8 mg mL⁻¹ (Supplementary Fig. 17). The acquisition of SWNT and DWNT can be attributed to the reduced amount of carbon deposited on the surface of the Fe catalyst. The existence of radial breathing mode (RBM) stretching vibration peak (100-300 cm⁻¹) in Raman spectrum also proved the synthesis of SWNT (Supplementary Figs. 17d and 17e). When the lignin concentration is higher than 5.5 mg mL⁻¹, the excessive lignin concentration will cause too many wall layers of CNTs, and a large number of carbon nanorods and amorphous carbon spheres also can be formed (Supplementary Figs. 17j-l).

Supplementary Figure 17. Effect of lignin concentration on the morphologies of CNTs.

(a) TEM image, (d) Raman spectrum, (g) growth mechanism of the CNTs prepared with lignin concentration of 0.4 mg mL^{-1} . (b) TEM image, (e) Raman spectrum, (h) growth mechanism of the CNTs prepared with lignin concentration of 0.8 mg mL^{-1} . (c) TEM image, (f) Raman spectrum, (i) growth mechanism of the CNTs prepared with lignin concentration of 2.5 mg mL^{-1} . TEM images of the (j) CNT aggregates prepared with lignin concentration of 5.5 mg mL^{-1} , and enlarged images of (k) carbon nanorods and (l) amorphous carbon spheres.

Corresponding changes:

Page 13 in Manuscript:

We also investigated the effect of lignin concentration on the morphology of CNTs. The results show that single-walled carbon nanotubes (SWNT) and double-walled carbon nanotubes (DWNT) can be obtained when the lignin concentration is lower than 0.8 mg mL^{-1}

(Supplementary Fig. 17). The acquisition of SWNT and DWNT can be attributed to the reduced amount of carbon deposited on the surface of the Fe catalyst. The existence of radial breathing mode (RBM) stretching vibration peak ($100\text{-}300\text{ cm}^{-1}$) in Raman spectrum also proved the synthesis of SWNT (Supplementary Figs. 17d-e). When the lignin concentration is higher than 5.5 mg mL^{-1} , the excessive lignin will cause too many wall layers of CNTs, and a large number of carbon nanorods and amorphous carbon spheres also can be formed (Supplementary Figs. 17j-l).

Comment 4: Scale bar for Figure 3a is required.

Response: We thank the reviewer for the comments.

According to the suggestion of the reviewer, we added the scale bar in Fig. 3a in the revised manuscript.

Corresponding changes:

Page 11 in Manuscript:

Fig. 3 Synthesis and structures of the lignin-derived CNTs.

Comment 5: *The cross-sections in Figure S17c and d were obtained from distorted surfaces/areas and, therefore, should not be used to determine the density of the fiber structure.*

Response: We thank the reviewer for the comments.

In Supplementary Fig. 17 in the original Supplementary Information, we provided the cross-sectional SEM images of the TCFs and RCFs to intuitively compare the internal microstructure of the CNT fibers obtained by two different densification methods, rather than calculate the density of the CNT fibers.

The density of the CNT fibers is calculated based on their cross-sectional area. The cross-section of the TCFs and RCFs are approximately circular and rectangular, respectively (Supplementary Fig. 20). The densities of the TCFs and RCFs are 0.64 g cm^{-3} and 1.49 g cm^{-3} , respectively, based on the mass and volume of the fibers.

Supplementary Figure 20. (a) SEM image and (b) calculation model of the TCFs. (c, d) SEM images and (e) calculation model of the RCFs.

Corresponding changes:

Page 14 in Manuscript:

As can be seen from the cross-section of the TCFs, the inside of the fibers is not dense enough (Supplementary Figs. 19c and 19d), which results in a low density of 0.64 g cm^{-3} (Supplementary Figs. 20a-b).

Page 14 in Manuscript:

Compared with twisting, CNT fibers prepared by rolling (rolled CNT fibers, RCFs) have a denser structure (Fig. 4d and Supplementary Fig. 19e) and improved fiber orientation (Supplementary Fig. 19f) due to the greater stress applied to the CNT fibers. The density of the RCFs is as high as 1.49 g cm^{-3} (Supplementary Figs. 20c-e).

Page 22 in Manuscript:

The density of the CNT fibers is calculated based on their mass and volume.

Comment 6: *The authors mentioned that the process had low energy consumption. How was the energy consumption estimated in this study? This should consider the post-treatment processes.*

Response: We thank the reviewer for the comments.

Since the energy consumption of CNT fibers in the preparation process is mainly concentrated in the high-temperature processing part, so only the energy consumed by lignin pyrolysis was considered in our original manuscript.

For the post-treatment process of CNT fibers, the energy consumption is mainly concentrated in fiber collection, twisting and rolling. An optical axis motor (2GN-18K-50K, rated power=6 W) from Taizhou Weichuang Electromechanical Equipment Co., Ltd. was used for fiber collection, and the energy consumption was 0.02 MJ h⁻¹. A yarn twist meter (Y331A, rated power≤25 W) from Changzhou Yifangyi Spinning Instrument Co., Ltd. was used for fiber twisting, and the energy consumption was 0.09 MJ h⁻¹. A small automatic roll-to-roll machine (MSK-HRP-04-RD, maximum power=20 W) from Hefei Kejing Material Technology Co., Ltd. was used for fiber rolling, and the energy consumption was 0.07 MJ h⁻¹ (Supplementary Fig. 24).

Based on the energy consumption of CNT synthesis and post-processing of CNT fibers, the energy consumption per unit length of CNT fibers is 0.12 MJ, which is significantly lower than that of carbon fibers per unit length (>0.22 MJ) (Supplementary Fig. 24). Therefore, we claimed that our process for preparing CNT fibers has a low energy consumption.

Corresponding changes:

Page 39 in Supplementary Information:

Note: For the preparation of carbon fibers, the high-temperature processing part includes pretreatment, carbonization and graphitization, and the data of its energy consumption referred to the research report of Deakin University^{59,60}. For the preparation of our CNT fibers, the energy consumption is mainly in the pyrolysis part of lignin. The pyrolysis of lignin is carried out in a tubular furnace (GSL-1400X, Hefei kejing Material Technology Co., Ltd., China) with a power of 4 KW, which consumes at most 14.4 MJ of energy per hour. For the post-treatment process of CNT fibers, the energy consumption is mainly concentrated in fiber collection, twisting and rolling. An optical axis motor (2GN-18K-50K, rated power=6 W) from Taizhou Weichuang Electromechanical Equipment Co., Ltd. was used for fiber collection, and the

energy consumption was 0.02 MJ h⁻¹. A yarn twist meter (Y331A, rated power≤25 W) from Changzhou Yifangyi Spinning Instrument Co., Ltd. was used for fiber twisting, and the energy consumption was 0.09 MJ h⁻¹. A small automatic roll-to-roll machine (MSK-HRP-04-RD, maximum power=20 W) from Hefei Kejing Material Technology Co., Ltd. was used for fiber rolling, and the energy consumption was 0.07 MJ h⁻¹.

Supplementary Figure 24. Comparison of energy consumption between lignin-based CNT fibers prepared by FCCVD method in our work and lignin-based carbon fibers prepared by conventional spinning method.

Comment 7: How was the porosity of the twisted and rolled fibers determined?

Response: We thank the reviewer for the comments.

The porosity of CNT fibers is calculated based on the density of CNT fibers and pure CNT materials. The densities of the TCFs and RCFs are 0.64 g cm⁻³ and 1.49 g cm⁻³, respectively. The density of pure CNT materials is 2.1 g cm⁻³. According to the following equation, the porosities of TCFs and RCFs are calculated as 69.5% and 29%, respectively.

$$\varphi(\%) = \frac{\rho_{\text{CNT}} - \rho_{\text{CNT fiber}}}{\rho_{\text{CNT}}} \times 100\%$$

Corresponding changes:

Page 22 in Manuscript:

The density of the CNT fibers is calculated based on their mass and volume. The porosity (φ) of the CNT fibers is calculated according to the following equation: $\varphi(\%) =$

$\frac{\rho_{\text{CNT}} - \rho_{\text{CNT fiber}}}{\rho_{\text{CNT}}} \times 100\%$, where ρ_{CNT} and $\rho_{\text{CNT fiber}}$ are the densities of pure CNT ($\rho_{\text{CNT}}=2.1 \text{ g cm}^{-3}$) and CNT fibers, respectively.

Reference:

1. Lu, Q., et al. Determination of carbon nanotube density by gradient sedimentation. *J. Phys. Chem. B* **110**, 24371-24376 (2006).

Reviewer #2

Lignin is a bulk by-product of paper industry, whereas has not been available used so far. In this work, high-performance CNT fibers were prepared by using industrial lignin as the carbon source under a highly continuous production speed. This paper was well written and the data was sufficient. This work would provide a new idea for the high-value utilization of industrial lignin or other biomass-based materials. However, some parts within the manuscript still needed to be improved. I think that it could be acceptable in Nature Communications after addressing the following comments:

Response: We thank the reviewer for the positive comments.

Comment 1: *The authors have provided the structural characterizations of the raw lignin, as well as the effects of functional groups and chemical bonds on the pyrolysis of lignin. However, I still wonder whether there are C-C bonds between the aromatic rings of the raw lignin?*

Response: We thank the reviewer for the comments.

According to literature research, the existence of C-C bonds between aromatic rings of lignin has been proved in some studies^{1,2}. However, according to our NMR results, we did not find C-C bonds between aromatic rings of the raw lignin, which may be attributed to the low content of C-C bonds in the lignin we used.

References:

1. Capanema, E. A., Balakshin, M. Y. & Kadla, J. F. A comprehensive approach for quantitative lignin characterization by NMR spectroscopy. *J Agric Food Chem* **52**, 1850-1860 (2004).
2. Karhunen, P., Rummakko, P., Sipila, J., Brunow, G. & Kilpelainen, I. The formation of dibenzodioxocin structures by oxidative coupling—a model reaction for lignin biosynthesis. *Tetrahedron Lett.* **36**, 4501-4504 (1995).

Comment 2: *This work investigated the utilization of lignin as a carbon source for the synthesis of CNTs. The carbon atoms existed in the side chain, benzene ring and methoxy groups of lignin macromolecules. So what structures in lignin does H₂O, H₂, CO, CH₄ and CO₂ derived from? Why is the yield of CO greater than that of CO₂? The author should give a specific explanation in the manuscript.*

Response: We thank the reviewer for the comments.

CO mainly comes from the cleavage of ether bonds in the side chains and between the aromatic rings in lignin, as well as the secondary decomposition of some volatiles. CO₂ mainly

comes from the cleavage and reformation of reactive functional groups (such as carbonyl and carboxyl groups) in the side chains. CH₄ is derived from the side chain cleavage and demethylation of methoxy groups on the benzene rings ¹. H₂O is mainly produced by the hydroxyl groups on the aliphatic side chains of lignin ². The formation of H₂ can be attributed to the rearrangement of broken bonds in the aromatic rings ³.

CO is an effective carbon source for the synthesis of CNTs ⁴, and its content is significantly higher than that of CH₄ and CO₂. The high yield of CO in lignin pyrolysis products can be attributed to its wide range of sources, including cleavage of the ether bonds in the side chains, cleavage of the ether bonds between the aromatic rings and the secondary decomposition of some volatiles.

Corresponding changes:

Page 7 in Manuscript:

CO mainly comes from the cleavage of ether bonds in the side chains and between the aromatic rings in lignin, as well as the secondary decomposition of some volatiles. CO₂ mainly comes from the cleavage and reformation of reactive functional groups (such as carbonyl and carboxyl groups) in the side chains. CH₄ is derived from the side chain cleavage and demethylation of methoxy groups on the benzene rings ³⁵. H₂O is mainly produced by the hydroxyl groups on the aliphatic side chains of lignin ³⁶. The formation of H₂ can be attributed to the rearrangement of broken bonds in the aromatic rings ³⁷. CO is an effective carbon source for the synthesis of CNTs ³⁸, and its content is significantly higher than that of CH₄ and CO₂ due to the wide range of sources.

References:

1. Wang, S. et al. Comparison of the pyrolysis behavior of lignins from different tree species. *Biotechnol. Adv.* **27**, 562-567 (2009).
2. Nunn, T. R., Howard, J. B., Longwell, J. P. & Peters, W. A. Product compositions and kinetics in the rapid pyrolysis of sweet gum hardwood. *Ind. Eng. Chem. Process Des. Dev.* **24**, 836-844 (1985).
3. Avni, E., Coughlin, R. W., Solomon, P. R. & King, H. H. Mathematical modelling of lignin pyrolysis. *Fuel* **64**, 1495-1501 (1985).
4. Nasibulin, A. G., Pikhitsa, P. V., Jiang, H. & Kauppinen, E. I. Correlation between catalyst particle and single-walled carbon nanotube diameters. *Carbon* **43**, 2251-2257 (2005).

Comment 3: *How do the authors calculate the purity of the generated CNTs?*

Response: We thank the reviewer for the comments.

The purity of the generated CNTs was calculated based on the thermogravimetric analysis data of CNT fibers (Fig. 3h). The original CNT fibers are mainly composed of C and Fe elements (Figs. 3l-n). In thermogravimetric analysis, C was completely removed from the sample, and only Fe was remained. In fact, Fe in the sample was converted to Fe_2O_3 when heated at high temperature in air¹, so the residue of the sample in Fig. 3h was Fe_2O_3 . We recalculated the Fe content in the sample by removing oxygen. The actual Fe content (impurity content) in the sample is 17.3%. Therefore, the CNT mass fraction in the sample should be 82.7%.

Fig. 3 Synthesis and structures of the lignin-derived CNTs. (a) Digital image and (b) SEM image of a CNT sock. (c) SEM image and (d) diameter distribution of the CNTs. (e, f) TEM images of the CNTs. (g) Raman spectrum and (h) thermogravimetric analysis (TGA) of the CNTs. (i) TEM image of the CNTs and catalyst particles. (j) Lattices of the CNTs and iron catalysts. (k) Schematic showing the growth mechanism of the CNTs on iron catalysts. (l) XRD pattern of the CNTs. (m) Elemental mapping images of the CNTs. (n) TEM-EDS image of the CNTs and the element contents.

Corresponding changes:

Page 9 in Manuscript:

TGA result shows that the mass fraction of the CNTs in the aggregates is 82.7% (Fig. 3h), which is similar to the purity of the CNTs prepared by the same method⁴⁵. Note that Fe in the sample was converted to Fe₂O₃ when heated at high temperature in air, so the removal of oxygen in Fe₂O₃ is required to calculate the impurity content⁴⁶. In addition, based on the carbon content (61.9%) and feeding rate (4.8 mg min⁻¹) of lignin as well as the preparation rate (1.46 mg min⁻¹) and purity (82.7%) of the CNT aggregates, the yield of the CNTs is about 40.6%.

Reference:

1. Zhan, H., Chen, Y. W., Shi, Q. Q., Zhang, Y., Mo, R. W. & Wang, J. N. Highly aligned and densified carbon nanotube films with superior thermal conductivity and mechanical strength. *Carbon* **186**, 205-214 (2022).

Comment 4: *The authors have mentioned that the rolling method can make fibers denser than the twisting method. The author should explain the corresponding mechanism in the text.*

Response: We thank the reviewer for the comments.

The rolling method can apply greater stress to the CNT fibers, and the CNTs in the fibers are closely arranged, resulting in greater fiber density (the density of RCFs is 1.49 g cm⁻³). As the friction between CNTs in the RCFs increases, and the slippage between CNTs becomes more difficult, thus significantly improving the mechanical properties of the fibers. However, for twisting method, too much twisting force will cause fiber fracture. Therefore, compared with rolling method, the force exerted by twisting on CNT fibers is smaller, and the resultant CNT fibers are loosely stacked with a density of only 0.64 g cm⁻³, so the mechanical properties of the TCFs are also poor.

The corresponding mechanism about the rolling method can make fibers denser than the twisting method have been added in the revised manuscript and highlighted in blue.

Corresponding changes:

Page 14 in Manuscript:

Compared with twisting, CNT fibers prepared by rolling (rolled CNT fibers, RCFs) have a denser structure (Fig. 4d and Supplementary Fig. 19e) and improved fiber orientation (Supplementary Fig. 19f) due to the greater stress applied to the CNT fibers.

Comment 5: *Various amounts of lignin were applied in the synthesis of CNTs. So what is the effect of these variables on the resulting CNTs? Furthermore, how does the injection rate (1-10 mL min⁻¹) affect on products?*

Response: We thank the reviewer for the comments.

We investigated the effect of lignin concentration on the morphology of CNTs. The results show that single-walled carbon nanotubes (SWNT) and double-walled carbon nanotubes (DWNT) can be obtained when the lignin concentration is lower than 0.8 mg mL⁻¹ (Supplementary Fig. 17). The acquisition of SWNT and DWNT can be attributed to the reduced amount of carbon deposited on the surface of the Fe catalyst. The existence of radial breathing mode (RBM) stretching vibration peak (100-300 cm⁻¹) in Raman spectrum also proved the synthesis of SWNT (Supplementary Figs. 17d-e). When the lignin concentration is higher than 5.5 mg mL⁻¹, the excessive lignin will cause too many wall layers of CNTs, and a large number of carbon nanorods and amorphous carbon spheres also can be formed (Supplementary Figs. 17j-l).

We also studied the effect of injection rate of lignin solution (1-10 mL min⁻¹) on the CNT fiber preparation. When the injection rate is lower than 1.5 mL min⁻¹, it's difficult to observe solid formation in the tubular furnace. When the injection rate is in the range of 1.5-2.5 mL min⁻¹, a small amount of CNT aerogels can be synthesized, but CNT fibers cannot be continuously prepared. The optimal injection rate range is 2.5-4.5 mL min⁻¹, in which CNT fibers can be continuously prepared. When the injection rate is higher than 4.5 mL min⁻¹, it's easy for the lignin solution to form aggregation and spray flame, which makes CNT fiber preparation unstable.

Supplementary Figure 17. Effect of lignin concentration on the morphologies of CNTs.

(a) TEM image, (d) Raman spectrum, (g) growth mechanism of the CNTs prepared with lignin concentration of 0.4 mg mL^{-1} . (b) TEM image, (e) Raman spectrum, (h) growth mechanism of the CNTs prepared with lignin concentration of 0.8 mg mL^{-1} . (c) TEM image, (f) Raman spectrum, (i) growth mechanism of the CNTs prepared with lignin concentration of 2.5 mg mL^{-1} . TEM images of the (j) CNT aggregates prepared with lignin concentration of 5.5 mg mL^{-1} , and enlarged images of (k) carbon nanorods and (l) amorphous carbon spheres.

Corresponding changes:

Page 13 in Manuscript:

We also investigated the effect of lignin concentration on the morphology of CNTs. The results show that single-walled carbon nanotubes (SWNT) and double-walled carbon nanotubes (DWNT) can be obtained when the lignin concentration is lower than 0.8 mg mL^{-1}

(Supplementary Fig. 17). The acquisition of SWNT and DWNT can be attributed to the reduced amount of carbon deposited on the surface of the Fe catalyst. The existence of radial breathing mode (RBM) stretching vibration peak ($100\text{-}300\text{ cm}^{-1}$) in Raman spectrum also proved the synthesis of SWNT (Supplementary Figs. 17d-e). When the lignin concentration is higher than 5.5 mg mL^{-1} , the excessive lignin will cause too many wall layers of CNTs, and a large number of carbon nanorods and amorphous carbon spheres also can be formed (Supplementary Figs. 17j-l).

Page 13 in Manuscript:

The effect of injection rate of lignin solution ($1\text{-}10\text{ mL min}^{-1}$) on the CNT fiber preparation was further studied. When the injection rate is lower than 1.5 mL min^{-1} , it's difficult to observe solid formation in the tubular furnace. When the injection rate is in the range of $1.5\text{-}2.5\text{ mL min}^{-1}$, a small amount of CNT aerogels can be synthesized, but CNT fibers cannot be continuously prepared. The optimal injection rate range is $2.5\text{-}4.5\text{ mL min}^{-1}$, in which CNT fibers can be continuously prepared. When the injection rate is higher than 4.5 mL min^{-1} , it's easy for the lignin solution to form aggregation and spray flame, which makes CNT fiber preparation unstable.

Comment 6: *The authors used methanol as a solvent, whether the lignin can be completely dissolved in methanol? The authors should provide the data in the manuscript.*

Response: We thank the reviewer for the comments.

Whether lignin can be completely dissolved in methanol depends on the solution concentration. When the concentration of lignin solutions is less than 1.5 mg mL^{-1} , lignin can be completely dissolved in methanol. No lignin precipitate can be found in these solutions after standing at room temperature for 12 h (Supplementary Fig. 18a). These low-concentration lignin solutions can be injected directly into the tubular furnace for CNT synthesis. When the solution concentration increases to more than 1.5 mg mL^{-1} , lignin cannot be completely dissolved. After standing for 12 h at room temperature, lignin precipitates from the solutions obtained by magnetic stirring (Supplementary Fig. 18b). The amount of lignin precipitation depends on the solution concentration, and the higher the concentration, the more lignin is precipitated. For these high-concentration lignin solutions, they should be continuously oscillated to keep them in a uniform dispersion state during the process of CNT synthesis (Supplementary Fig. 18c).

Supplementary Figure 18. Digital images of lignin solutions with a concentration of (a) 1.5 mg mL^{-1} , and (b) 6 mg mL^{-1} . (c) Schematic showing the oscillation of high-concentration lignin solutions during CNT synthesis.

Corresponding changes:

Page 13 in Manuscript:

In the process of CNT synthesis, the way lignin solution is injected into the tubular furnace depends on the solution concentration. When the concentration of lignin solutions is less than 1.5 mg mL^{-1} , lignin can be completely dissolved in methanol. No lignin precipitate can be found in these solutions after standing at room temperature for 12 h (Supplementary Fig. 18a). These low-concentration lignin solutions can be injected directly into the tubular furnace for CNT synthesis. When the solution concentration increases to more than 1.5 mg mL^{-1} , lignin cannot be completely dissolved. After standing for 12 h at room temperature, lignin precipitates from the solutions obtained by magnetic stirring (Supplementary Fig. 18b). The amount of lignin precipitation depends on the solution concentration, and the higher the concentration, the more lignin is precipitated. For these high-concentration lignin solutions, they should be continuously oscillated to keep them in a uniform dispersion state during the process of CNT synthesis (Supplementary Fig. 18c).

Comment 7: *In the “test performance” section, what is the length-diameter ratio of CNTs fibers used for the mechanical strength test? What is the effect of different length-diameter ratios on mechanical strength?*

Response: We thank the reviewer for the comments.

The cross-section of the TCFs is circular, and the diameter is about $38.05 \text{ }\mu\text{m}$. The length of TCFs used for mechanical property determination is 10 mm (initial gauge length), so the length-diameter ratio of TCFs is 263. RCFs have a rectangular cross-section. The length of TCFs is 10 mm, the width and thickness of the cross-section is $70.13 \text{ }\mu\text{m}$ and $7.01 \text{ }\mu\text{m}$, respectively, so the length-width ratio and length-thickness ratio of RCFs are 143 and 1427, respectively.

Previous studies have shown that the length-diameter ratio (that is gauge length) of CNT fibers has little influence on their mechanical strength¹. Therefore, a moderate length-diameter ratio of 10 mm was selected for the mechanical property determination of the CNT fibers in our study. Note that a gauge length of 10 mm is often used to test the mechanical properties of CNT fibers^{2,3,4}.

Corresponding changes:

Page 22 in Manuscript:

The mechanical properties of the CNT fibers were determined using a universal tensile testing machine (YG-004, Dahua Electronic, China) **and the gauge length was set as 10 mm.**

References:

1. Wang, J. N., Luo, X. G., Wu, T. & Chen, Y. High-strength carbon nanotube fibre-like ribbon with high ductility and high electrical conductivity. *Nat. Commun.* **5**, 3848 (2014).
2. Jiang, X. et al. Understanding the influence of single-walled carbon nanotube dispersion states on the microstructure and mechanical properties of wet-spun fibers. *Carbon* **169**, 17-24 (2020).
3. Shang, Y., Wang, Y., Li, S., Hua, C., Zou, M. & Cao, A. High-strength carbon nanotube fibers by twist-induced self-strengthening. *Carbon* **119**, 47-55 (2017).
4. Zhang, Y., Zheng, L., Sun, G., Zhan, Z. & Liao, K. Failure mechanisms of carbon nanotube fibers under different strain rates. *Carbon* **50**, 2887-2893 (2012).

Comment 8: *Other important physical parameters such as density, mass per unit length, elastic modulus and elongation at break should be given in the manuscript.*

Response: We thank the reviewer for the comments.

Per the suggestion of the reviewer, we added other important physical parameters, including density, mass per unit length, elastic modulus and elongation at break in the revised manuscript.

1) Density

The density of the CNT fibers is calculated based on their cross-sectional area. The cross-section of the TCFs and RCFs are approximately circular and rectangular, respectively (Supplementary Fig. 20). The densities of the TCFs and RCFs are 0.64 g cm⁻³ and 1.49 g cm⁻³, respectively, based on the mass and volume of the fibers.

Supplementary Figure 20. (a) SEM image and (b) calculation model of the TCFs. (c, d) SEM images and (e) calculation model of the RCFs.

2) Mass per unit length

The mass per unit length is calculated based on the mass and length of the CNT fibers, and the value for RCFs is 0.73 mg m^{-1} .

3) Elastic modulus

The elastic moduli of TCFs and RCFs, calculated from the slope of their tensile stress-strain, are $10.45 \pm 1.24 \text{ GPa}$ and $37.45 \pm 7.47 \text{ GPa}$, respectively.

4) Elongation at break

According to the tensile stress-strain curves of TCFs and RCFs, their elongation at break are $6.12 \pm 0.43\%$ and $5.62 \pm 0.18\%$, respectively.

Corresponding changes:

Page 14 in Manuscript:

As can be seen from the cross-section of the TCFs, the inside of the fibers is not dense enough (Supplementary Figs. 19c and 19d), which results in a low density of 0.64 g cm^{-3} (Supplementary Figs. 20a-b).

Page 14 in Manuscript:

Compared with twisting, CNT fibers prepared by rolling (rolled CNT fibers, RCFs) have a denser structure (Fig. 4d and Supplementary Fig. 19e) and improved fiber orientation (Supplementary Fig. 19f) due to the greater stress applied to the CNT fibers. The density of the RCFs is as high as 1.49 g cm^{-3} (Supplementary Figs. 20c-e).

Page 16 in Manuscript:

The tensile strength of TCFs and RCFs are $0.27 \pm 0.02 \text{ GPa}$ and $1.33 \pm 0.08 \text{ GPa}$, respectively, and their elastic moduli are $10.45 \pm 1.24 \text{ GPa}$ and $37.45 \pm 7.47 \text{ GPa}$, respectively

(Fig. 4g). RCFs have a denser structure and a more oriented structure compared to TCFs, which results in higher friction and more difficult slippage between CNTs in the fibers, thus achieving significantly better mechanical properties. The elongation at break of TCFs and RCFs are $6.12\pm 0.43\%$ and $5.62\pm 0.18\%$, respectively. Although TCFs and RCFs have similar elongation at break, RCFs exhibit significantly higher fracture work due to their significantly higher mechanical strength.

Page 22 in Manuscript:

The mass per unit length is calculated based on the mass and length of the CNT fibers, and the value for RCFs is 0.73 mg m^{-1} .

Comment 9: *In the synthesis process, the author used a large number of catalysts based on the quality of raw lignin. How to control production cost? How are these catalysts disposed after the reaction?*

Response: We thank the reviewer for the comments.

Suitable catalyst concentration is very important for the continuous preparation of CNT fibers. In our work, we use ferrocene as the catalyst, and the concentration of ferrocene in the lignin solutions is 0.005 g mL^{-1} , lower than that used in many literatures for the preparation of CNTs by similar methods (Supplementary Table 9).

In order to further control the production cost of CNT fibers, the following aspects can be considered: 1) Optimize the feeding method of lignin solutions, and explore the use of spray solution supply method to improve the efficiency of catalyst; 2) Adjust the flow rate of carrier gas to prolong the residence time of catalyst in the reaction zone, thus improving the catalytic efficiency and reducing the amount of catalyst; 3) Reduce the amount of catalyst by optimizing the proportion of each component in the lignin solutions.

Some Fe impurities will remain in the CNT fibers after the synthesis reaction. At present, there are two commonly used methods to remove these impurities. 1) Dissolving CNTs in chlorosulfonic acid to remove Fe impurities^{1,2}. 2) Heat treatment of CNTs at temperatures higher than 1500°C in an inert atmosphere or under vacuum conditions^{3,4}.

Corresponding changes:

Page 19 in Manuscript:

Although the method for improving the mechanical properties of CNT fibers by acid treatment and heat treatment have been reported ^{45, 83}, the additional processes inevitably increase the cost of fiber manufacturing and reduce the productivity, and are not conducive to the continuous preparation of CNT fibers.

Suitable catalyst concentration is very important for the continuous preparation of CNT fibers. In our work, we use ferrocene as the catalyst, and the concentration of ferrocene in the lignin solutions is 0.005 g mL⁻¹, lower than that used in many literatures for the preparation of CNTs by similar methods (Supplementary Table 9). In addition, the amount of catalyst is also very important to control the production cost of CNT fibers. In order to further control the production cost of CNT fibers, the following aspects can be considered: 1) Optimize the feeding method of lignin solutions, and explore the use of spray solution supply method to improve the efficiency of catalyst; 2) Adjust the flow rate of carrier gas to prolong the residence time of catalyst in the reaction zone, thus improving the catalytic efficiency and reducing the amount of catalyst; 3) Reduce the amount of catalyst by optimizing the proportion of each component in the lignin solutions.

Page 12 in Supplementary Information:

Supplementary Table 9. Comparison of the amount of ferrocene we used for preparing lignin-based CNT fibers with other literatures.

Carbon source (Carrier gas)	Ferrocene concentration in solution (g mL ⁻¹)	Mass fraction of ferrocene to carbon source (%)	Reference
Lignin (Ar)	0.005	0.62	Our work
N-hexane (H ₂ +Ar)	0.007-0.013	-	50
Xylene+ dichlorobenzene (H ₂)	0.1	-	51
Xylene (H ₂ +Ar)	0.05	-	52
Cyclohexane (H ₂)	0.02	-	53
Toluene (H ₂ +Ar)	0.03	-	54
Ethanol (H ₂ +Ar)	-	0.1-3	55
Ethanol (H ₂ +N ₂)	-	0.25-0.4	56
Toluene (H ₂)	-	1-9.6	57

References:

1. Tsentelovich, D. E. et al. Influence of carbon nanotube characteristics on macroscopic fiber properties. *ACS Appl. Mater. Interfaces* **9**, 36189-36198 (2017).
2. Taylor, L. W. et al. Improved properties, increased production, and the path to broad adoption of carbon nanotube fibers. *Carbon* **171**, 689-694 (2021).
3. Huang, W. Wang, Y. & Luo, G., Wei F. 99.9% purity multi-walled carbon nanotubes by vacuum high-temperature annealing. *Carbon* **41**, 2585-2590 (2003).
4. Andrews, R., Jacques, D., Qian, D. & Dickey, E. Purification and structural annealing of multiwalled carbon nanotubes at graphitization temperatures. *Carbon* **39**, 1681-1687 (2001).

Comment 10: *The author only compared the literatures of the biomass-derived carbon fiber materials. Comparisons with CNTs fibers generated by classical FCCVD methods should also be considered to support the superiority of this work.*

Response: We thank the reviewer for the comments.

According to the suggestion of the reviewer, we compared the mechanical strength and electrical conductivity of our CNT fibers with the reported biomass-derived carbon fibers, array CNT fibers, CNT fibers from FCCVD and wet-spun CNT fibers, as well as commercial carbon fibers and common metal materials.

Corresponding changes:

Page 15 in Manuscript:

Fig. 4 Preparation and properties of lignin-derived CNT fibers. Preparation diagrams of (a) TCFs and (b) RCFs. SEM images of (c) TCFs and (d) RCFs. Polarised Raman spectra of (e) TCFs and (f) RCFs. (g) Tensile stress-strain curves of the TCFs and RCFs. (h) Comparison of tensile strength and fracture work for the TCFs and RCFs. (i) Comparison of the thermal conductivity and density between our CNT films and other conductive materials ^{46, 52, 53, 54, 55, 56, 57, 58, 59}. (j) Comparison of the tensile strength and electrical conductivity of our CNT fibers with commercial carbon fibers ⁶⁰, wet-spun CNT fibers ^{18, 61, 62, 63}, CNT fibers from FCCVD ^{25, 64, 65, 66, 67, 68}, CNT fibers from array spinning ^{69, 70, 71, 72, 73}, common metal materials ⁵⁹ and biomass-derived carbon fibers ^{74, 75, 76, 77, 78, 79, 80}.

Page 17 in Manuscript:

We also proved that our CNT fibers have high electrical conductivity, and the electrical conductivity of the CNT fibers with a density of 1.49 g cm^{-3} is as high as $(6.03 \pm 0.25) \times 10^5 \text{ S m}^{-1}$, which is similar to that of the CNT fibers with similar structures prepared by the similar

method ⁶⁷. The electrical conductivity of our CNT fibers is higher than that of almost all reported biomass-derived carbon fibers and array CNT fibers as well as most commercial carbon fibers (Fig. 4j). It is worth noting that the electrical conductivity of our CNT fibers is lower than that of most wet-spun CNT fibers, which may be due to the higher purity and crystallinity of the CNTs used for wet-spinning as well as the higher density of the resultant CNT fibers (Fig. 4j and Supplementary Table 7).

Although the mechanical strength of the prepared CNT fibers is not yet comparable to that of commercial carbon fibers, it is higher than or similar to that of most reported biomass-derived carbon fibers, array CNT fibers, CNT fibers from FCCVD and wet-spun CNT fibers, as well as all common metal materials (Fig. 4j and Supplementary Table 7). It should be emphasized that the mechanical strength of our CNT fibers exceeds that of most CNT fibers prepared with fine chemicals (such as alkanes and aromatic hydrocarbons) as carbon sources. Taken together, our lignin-derived CNT fibers show the unprecedented integration of high tensile strength, thermal conductivity, and electrical conductivity, as well as continuous preparation process.

Page 9 in Supplementary Information:

Supplementary Table 7. Comparison of tensile strength and electrical conductivity of lignin-based CNT fibers with other carbon-based fibers and metal materials.

Materials	Tensile strength (GPa)	Conductivity (S m⁻¹)	Reference
Lignin-CNT fibers	1.33	6.03×10 ⁵	Our work
Biomass-derived carbon fibers	0.088	1.03×10 ⁴	17
Biomass-derived carbon fibers	0.369	1.91×10 ⁴	18
Biomass-derived carbon fibers	0.351	1.41×10 ⁴	19
Biomass-derived carbon fibers	0.129	5×10 ³	20
Biomass-derived carbon fibers	1.648	1.85×10 ⁴	21
Biomass-derived carbon fibers	0.763	2×10 ³	22
Biomass-derived carbon fibers	0.59	1×10 ³	23
Biomass-derived carbon fibers	0.57	3×10 ³	23
Array CNT fibers	0.656	4.08×10 ⁴	24

Array CNT fibers	1.408	5.84×10^4	25
Array CNT fibers	1.408	2.39×10^5	25
Array CNT fibers	1.90	6×10^4	26
Array CNT fibers	1.91	4.1×10^4	27
Array CNT fibers	0.389	1.05×10^5	28
FCCVD CNT fibers	1.0	1.43×10^5	29
FCCVD CNT fibers	0.36	2.0×10^5	30
FCCVD CNT fibers	0.38	4.6×10^5	31
FCCVD CNT fibers	0.32	2.0×10^6	32
FCCVD CNT fibers	0.36	1.27×10^5	33
FCCVD CNT fibers	4.34	2.05×10^6	33
FCCVD CNT fibers	0.27	1.657×10^5	34
FCCVD CNT fibers	2.81	1.2×10^6	34
Wet-spun CNT fibers	0.116	5×10^5	35
Wet-spun CNT fibers	1.0	2.9×10^6	36
Wet-spun CNT fibers	1.0	5.0×10^6	36
Wet-spun CNT fibers	2.4	8.5×10^6	37
Wet-spun CNT fibers	4.2	1.09×10^7	38
Hexcel (AS4)	4.27	6.5×10^4	39
Cytec (T300)	3.75	5.56×10^4	39
Toray (T300)	3.53	5.9×10^4	39
Toray (T1000G)	6.37	7.14×10^4	39
Toray (M55J)	4.02	1.25×10^5	39
Cytec (K-800X)	2.34	8.83×10^5	39
Cytec (K-1100)	3.10	9.09×10^5	39
Silver (Ag)	0.14	6.3×10^7	16
Copper (Cu)	0.21	5.8×10^7	16
Aluminum (Al)	0.1	3.5×10^7	16
Iron (Fe)	0.54	1.0×10^7	16

Reviewer #3

In this manuscript by Liu and co-workers, the authors investigated the possibility of synthesis of carbon nanotubes from kraft lignin solution in methanol. Based on this concept, a direct spinning variant of CVD was utilized to manufacture fibers in a continuous manner. The results are interesting, but some issues should be addressed first before the submission can be reconsidered for publication in Nature Communications. Please find suggestions below:

Response: We thank the reviewer for the positive comments.

Comment 1: *“The carbon sources used in this method are mainly from petroleum fine chemicals, such as methane, ethylene, ethanol and xylene” – aromatic solvents such as toluene should also be mentioned due to their widespread use.*

Response: We thank the reviewer for the comments.

According to the suggestion of the reviewer, we have supplemented “toluene” in the Introduction section in the revised manuscript.

Corresponding changes:

Page 4 in Manuscript:

The carbon sources used in this method are mainly from petroleum fine chemicals, such as methane, ethylene, ethanol, toluene and xylene²⁷.

Comment 2: *“After post-treatment, the lignin-based CNT fibers were endowed with a tensile strength of 1.35 GPa and an electrical conductivity of $6.28 \times 10^5 \text{ S m}^{-1}$. In addition, the continuous production of CNTs fibers from lignin with a 120 m h^{-1} production rate was achieved.” - these values should be compared with the whole state of the art (not just narrowed down to CNTs synthesized from natural resources but also from synthetic precursors such as alkanes and aromatic hydrocarbons). Such a summary would be useful to evaluate how good the reported values really are.*

Response: We thank the reviewer for the comments.

According to the suggestion of the reviewer, we compared the mechanical strength and electrical conductivity of our CNT fibers with the reported biomass-derived carbon fibers, array CNT fibers, CNT fibers from FCCVD and wet-spun CNT fibers, as well as commercial carbon fibers and common metal materials. In addition, we also provided a comparison of the production rates of our CNT fibers with those of other CNT fibers prepared from fine chemicals.

Corresponding changes:

Fig. 4 Preparation and properties of lignin-derived CNT fibers. Preparation diagrams of (a) TCFs and (b) RCFs. SEM images of (c) TCFs and (d) RCFs. Polarised Raman spectra of (e) TCFs and (f) RCFs. (g) Tensile stress-strain curves of the TCFs and RCFs. (h) Comparison of tensile strength and fracture work for the TCFs and RCFs. (i) Comparison of the thermal conductivity and density between our CNT films and other conductive materials ^{46, 52, 53, 54, 55, 56, 57, 58, 59}. (j) Comparison of the tensile strength and electrical conductivity of our CNT fibers with commercial carbon fibers ⁶⁰, wet-spun CNT fibers ^{18, 61, 62, 63}, CNT fibers from FCCVD ^{25, 64, 65, 66, 67, 68}, CNT fibers from array spinning ^{69, 70, 71, 72, 73}, common metal materials ⁵⁹ and biomass-derived carbon fibers ^{74, 75, 76, 77, 78, 79, 80}.

We also proved that our CNT fibers have high electrical conductivity, and the electrical conductivity of the CNT fibers with a density of 1.49 g cm^{-3} is as high as $(6.03 \pm 0.25) \times 10^5 \text{ S}$

m⁻¹, which is similar to that of the CNT fibers with similar structures prepared by the similar method ⁶⁷. The electrical conductivity of our CNT fibers is higher than that of almost all reported biomass-derived carbon fibers and array CNT fibers as well as most commercial carbon fibers (Fig. 4j). It is worth noting that the electrical conductivity of our CNT fibers is lower than that of most wet-spun CNT fibers, which may be due to the higher purity and crystallinity of the CNTs used for wet-spinning as well as the higher density of the resultant CNT fibers (Fig. 4j and Supplementary Table 7).

Although the mechanical strength of the prepared CNT fibers is not yet comparable to that of commercial carbon fibers, it is higher than or similar to that of most reported biomass-derived carbon fibers, array CNT fibers, CNT fibers from FCCVD and wet-spun CNT fibers, as well as all common metal materials (Fig. 4j and Supplementary Table 7). It should be emphasized that the mechanical strength of our CNT fibers exceeds that of most CNT fibers prepared with fine chemicals (such as alkanes and aromatic hydrocarbons) as carbon sources. Taken together, our lignin-derived CNT fibers show the unprecedented integration of high tensile strength, thermal conductivity, and electrical conductivity, as well as continuous preparation process.

Page 18 in Manuscript:

Compared with the preparation of CNT fibers using fine chemicals as raw materials, the preparation efficiency of our method is lower because it takes a certain amount of time for lignin to decompose into small molecules (Supplementary Table 8). However, the production of traditional lignin-based carbon fibers involves spinning and multi-step heat treatment. It takes at least 90 minutes to get lignin-based carbon fibers, and the fiber preparation rate is only 20-35 m h⁻¹ ⁸¹.

Page 9 in Supplementary Information:

Supplementary Table 7. Comparison of tensile strength and electrical conductivity of lignin-based CNT fibers with other carbon-based fibers and metal materials.

Materials	Tensile strength (GPa)	Conductivity (S m⁻¹)	Reference
Lignin-CNT fibers	1.33	6.03×10 ⁵	Our work
Biomass-derived carbon fibers	0.088	1.03×10 ⁴	17

Biomass-derived carbon fibers	0.369	1.91×10^4	18
Biomass-derived carbon fibers	0.351	1.41×10^4	19
Biomass-derived carbon fibers	0.129	5×10^3	20
Biomass-derived carbon fibers	1.648	1.85×10^4	21
Biomass-derived carbon fibers	0.763	2×10^3	22
Biomass-derived carbon fibers	0.59	1×10^3	23
Biomass-derived carbon fibers	0.57	3×10^3	23
Array CNT fibers	0.656	4.08×10^4	24
Array CNT fibers	1.408	5.84×10^4	25
Array CNT fibers	1.408	2.39×10^5	25
Array CNT fibers	1.90	6×10^4	26
Array CNT fibers	1.91	4.1×10^4	27
Array CNT fibers	0.389	1.05×10^5	28
FCCVD CNT fibers	1.0	1.43×10^5	29
FCCVD CNT fibers	0.36	2.0×10^5	30
FCCVD CNT fibers	0.38	4.6×10^5	31
FCCVD CNT fibers	0.32	2.0×10^6	32
FCCVD CNT fibers	0.36	1.27×10^5	33
FCCVD CNT fibers	4.34	2.05×10^6	33
FCCVD CNT fibers	0.27	1.657×10^5	34
FCCVD CNT fibers	2.81	1.2×10^6	34
Wet-spun CNT fibers	0.116	5×10^5	35
Wet-spun CNT fibers	1.0	2.9×10^6	36
Wet-spun CNT fibers	1.0	5.0×10^6	36
Wet-spun CNT fibers	2.4	8.5×10^6	37
Wet-spun CNT fibers	4.2	1.09×10^7	38
Hexcel (AS4)	4.27	6.5×10^4	39
Cytec (T300)	3.75	5.56×10^4	39
Toray (T300)	3.53	5.9×10^4	39
Toray (T1000G)	6.37	7.14×10^4	39

Toray (M55J)	4.02	1.25×10^5	39
Cytec (K-800X)	2.34	8.83×10^5	39
Cytec (K-1100)	3.10	9.09×10^5	39
Silver (Ag)	0.14	6.3×10^7	16
Copper (Cu)	0.21	5.8×10^7	16
Aluminum (Al)	0.1	3.5×10^7	16
Iron (Fe)	0.54	1.0×10^7	16

Page 11 in Supplementary Information:

Supplementary Table 8. Comparison of production rate of CNT fibers from lignin and fine chemicals.

Methods	Carbon source	Production rate (m h ⁻¹)	Reference	
FCCVD	Biomass	Lignin	120	Our work
Array		Acetylene	60-600	40
Wet-spinning		Methane	300-540	41
FCCVD		Ethanol	120-1200	33
FCCVD		Methane	300	42
FCCVD		Butanol	300-480	43
FCCVD	Fine	Acetone	300-1800	44
FCCVD	chemicals	Methane	330	45
FCCVD		Butanol	420-540	46
FCCVD		Acetone	450-540	47
FCCVD		Ethanol	600	48
FCCVD		Butanol	600	49
FCCVD		Toluene	900	31

Comment 3: “TGA result shows that the mass fraction of the CNTs in the aggregates is 75.3% (Figure 3h), which indicates that the lignin-based CNT fibers have high purity” – judging by the provided thermogram, the nanotubes are of poor crystallinity. Yet, the authors report extremely high electrical conductivity and thermal conductivity. Please comment on this issue.

Response: We thank the reviewer for the comments.

The original CNT fibers are mainly composed of C and Fe elements (Figs. 3l-n). In thermogravimetric analysis, C was completely removed from the sample, and only Fe was remained. In fact, Fe in the sample was converted to Fe_2O_3 when heated at high temperature in air¹, so the residue of the sample in Fig. 3h was Fe_2O_3 . We recalculated the Fe content in the sample by removing oxygen. The actual Fe content (impurity content) in the sample is 17.3%. Therefore, the CNT mass fraction in the sample should be 82.7%. In addition, the CNTs prepared using our method have an I_G/I_D of 3.84, indicating that they are well crystallized.

Fig. 3 Synthesis and structures of the lignin-derived CNTs. (a) Digital image and (b) SEM image of a CNT sock. (c) SEM image and (d) diameter distribution of the CNTs. (e, f) TEM images of the CNTs. (g) Raman spectrum and (h) thermogravimetric analysis (TGA) of the CNTs. (i) TEM image of the CNTs and catalyst particles. (j) Lattices of the CNTs and iron catalysts. (k) Schematic showing the growth mechanism of the CNTs on iron catalysts. (l) XRD pattern of the CNTs. (m) Elemental mapping images of the CNTs. (n) TEM-EDS image of the CNTs and the element contents.

The factors affecting the electrical conductivity and thermal conductivity of CNT fibers include CNT type (single-wall, double-wall and multi-wall), crystallinity (I_G/I_D), purity, density, fiber microstructure (aligned and random), etc. Although the CNTs we prepared are

multiwalled, they have a purity of 82.7% and an I_G/I_D of 3.84, which combines with the dense structure result in the high electrical conductivity and thermal conductivity. The electrical conductivity of the CNT fibers with a density of 1.49 g cm^{-3} is $6.03 \times 10^5 \text{ S m}^{-1}$, which is similar to that of the CNT fibers with similar structures prepared by the similar method (Table R1) ². Our CNT films with a density of 0.82 g cm^{-3} exhibit a thermal conductivity of $33.21 \pm 0.76 \text{ W m}^{-1} \text{ K}^{-1}$, which is close to that of the CNT films with similar characteristics prepared by the similar method (Table R2) ¹.

Table R1. Comparison of the electrical conductivity of the CNT fibers prepared by our method with that of other CNT fibers prepared by the similar method.

Method	Raw material	CNT type	I_G/I_D	Purity (%)	Density (g cm^{-3})	Post processing	Conductivity (S m^{-1})	Reference
FCCVD	Lignin	MWNT	3.84	82.7	1.49	Mechanical molding	6.03×10^5	Our work
FCCVD	Ethanol	MWNT	3.74	85-93	1.3-1.8	Mechanical molding	2.24×10^6	2

Table R2. Comparison of the thermal conductivity of the CNT fibers prepared by our method with that of other CNT fibers prepared by the similar method.

CNT type	Raw material	Film structure	I_G/I_D	Purity (%)	Density (g cm^{-3})	k ($\text{W m}^{-1} \text{ K}^{-1}$)	Measurement method	Reference
MWNT film	Lignin	Random	3.8	82.7	0.82	33.21	LFA	Our work
MWNT film	Ethanol	Random	5.2	93.5	0.37	20.91-44.46	LFA	1
MWNT film	Ethanol	Random	5.2	93.5	1.01	110.57-158.66	LFA	1
MWNT film	Ethanol	Random	5.2	93.5	1.59	362.55-458.58	LFA	1

Note: LFA: Laser flash analysis

Corresponding changes:

Page 9 in Manuscript:

TGA result shows that the mass fraction of the CNTs in the aggregates is **82.7%** (Fig. 3h),

which is similar to the purity of the CNTs prepared by the same method ⁴⁵. Note that Fe in the sample was converted to Fe₂O₃ when heated at high temperature in air, so the removal of oxygen in Fe₂O₃ is required to calculate the impurity content ⁴⁶. In addition, based on the carbon content (61.9%) and feeding rate (4.8 mg min⁻¹) of lignin as well as the preparation rate (1.46 mg min⁻¹) and purity (82.7%) of the CNT aggregates, the yield of the CNTs is about 40.6%.

Page 16 in Manuscript:

In addition to excellent mechanical properties, the CNT films with a density of 0.82 g cm⁻³ exhibit high thermal conductivity of 33.21±0.76 W m⁻¹ K⁻¹ (Supplementary Fig. 23). Compared to biomass-derived carbon materials (0.06-24 W m⁻¹ K⁻¹), our CNT films possess higher thermal conductivity, comparable to that of the CNT films with similar characteristics prepared by the similar method (20.91-458.58 W m⁻¹ K⁻¹) as well as some common metals (30-500 W m⁻¹ K⁻¹) (Fig. 4i and Supplementary Table 6). Considering that the CNT films have significantly lower density (0.82 g cm⁻³) than common metals (2.7-10.49 g cm⁻³), they can be used in some fields that require lightweight thermal conductive materials.

Page 17 in Manuscript:

We also proved that our CNT fibers have high electrical conductivity, and the electrical conductivity of the CNT fibers with a density of 1.49 g cm⁻³ is as high as (6.03±0.25)×10⁵ S m⁻¹, which is similar to that of the CNT fibers with similar structures prepared by the similar method ⁶⁷. The electrical conductivity of our CNT fibers is higher than that of almost all reported biomass-derived carbon fibers and array CNT fibers as well as most commercial carbon fibers (Fig. 4j). It is worth noting that the electrical conductivity of our CNT fibers is lower than that of most wet-spun CNT fibers, which may be due to the higher purity and crystallinity of the CNTs used for wet-spinning as well as the higher density of the resultant CNT fibers (Fig. 4j and Supplementary Table 7).

Page 8 in Supplementary Information:

Supplementary Table 6. Comparison of thermal conductivity of our CNT films and CNT films with similar characteristics prepared by the similar method, other biomass-derived carbon materials, as well as common metals.

Materials	Thermal conductivity (W m ⁻¹ K ⁻¹)	Density (g cm ⁻³)	Reference
-----------	--	-------------------------------	-----------

Lignin-CNT films	33.21	0.82	Our work
PMMA/CNT	3.44	1.18	8
Polycarbonate/CNT	1.27	1.2	9
Lignin-based carbon foams	0.75	0.68	10
Lignin-based carbon fibers	24	2.189	11
Lignin-based carbon fibers	1.8	2	12
Lignin wood	0.23	1.2	13
Lignin aerogels	0.06	2.5	14
Ethanol-CNT films	20.91-458.58	0.37-1.59	15
Silver (Ag)	419	10.49	16
Copper (Cu)	385	7.76	16
Aluminum (Al)	210	2.7	16
Iron (Fe)	76.2	7.87	16

References:

1. Zhan, H., Chen, Y. W., Shi, Q. Q., Zhang, Y., Mo, R. W. & Wang, J. N. Highly aligned and densified carbon nanotube films with superior thermal conductivity and mechanical strength. *Carbon* **186**, 205-214 (2022).
2. Wang, J. N., Luo, X. G., Wu, T. & Chen, Y. High-strength carbon nanotube fibre-like ribbon with high ductility and high electrical conductivity. *Nat. Commun.* **5**, 3848 (2014).

Comment 4: *Regarding the electrical conductivity of CNT fibers, a primary source of error, which may greatly affect the result, is the cross-section area. Because the authors report very high electrical conductivity values (on the order of thousands of S/cm), more information should be provided on how these values were obtained (especially how the diameter was established). Currently, the following description is not very informative “The determination of electrical conductivity was performed on a Digit Graphical Touchscreen Digital Multimeter (DMM6500 6½)”. Was it a two- or four-probe approach?*

Response: We thank the reviewer for the comments.

The electrical conductivity of the CNT fibers was measured by two-probe method. The distance between the two probes was set at 1 cm. The resistance of the CNT fibers was measured by a multimeter, as shown in Fig. R3a. The electrical conductivity (σ , S m^{-1}) is calculated by the following equation: $\sigma = \frac{L}{R \times S}$, where L is the distance between the two probes ($L=1$ cm), R is the resistance of the CNT fibers (Ω), and S is the cross-sectional area (m^2). The samples used for electrical conductivity test are RCFs, whose cross-section is rectangular (Figs. R3b-d). Five conductivity values are obtained and their average is presented, that is $(6.03 \pm 0.25) \times 10^5 \text{ S m}^{-1}$.

Figure R3. Electrical conductivity test of RCFs. (a) Digital images showing the resistance of the RCFs. (b, c) SEM images and (d) cross-sectional area calculation model of the RCFs.

Corresponding changes:

Page 17 in Manuscript:

We also proved that our CNT fibers have high electrical conductivity, and the electrical conductivity of the CNT fibers with a density of 1.49 g cm^{-3} is as high as $(6.03 \pm 0.25) \times 10^5 \text{ S m}^{-1}$, which is similar to that of the CNT fibers with similar structures prepared by the similar method⁶⁷. The electrical conductivity of our CNT fibers is higher than that of almost all reported biomass-derived carbon fibers and array CNT fibers as well as most commercial carbon fibers (Fig. 4j). It is worth noting that the electrical conductivity of our CNT fibers is lower than that of most wet-spun CNT fibers, which may be due to the higher purity and crystallinity of the CNTs used for wet-spinning as well as the higher density of the resultant CNT fibers (Fig. 4j and Supplementary Table 7).

Page 22 in Manuscript:

The electrical conductivity of the CNT fibers was measured by two-probe method. The distance between the two probes was set at 1 cm. The resistance of the CNT fibers was measured by a Digit Graphical Touchscreen Digital Multimeter (DMM6500 6½, Keithley, USA). The electrical conductivity (σ , S m⁻¹) is calculated by the following equation: $\sigma = \frac{L}{R \times S}$, where L is the distance between the two probes ($L=1$ cm), R is the resistance of the CNT fibers (Ω), and S is the cross-sectional area (m²). RCFs with a rectangular cross-section were used for the electrical conductivity determination. Five conductivity values were obtained and their average was presented.

Comment 5: *Whenever possible, error analysis should be conducted. The absence of error bars casts doubt about the statistical significance of the reported data.*

Response: We thank the reviewer for the comments.

According to the suggestion of the reviewer, error analysis has been conducted in the revised manuscript.

Corresponding changes:

Page 16 in Manuscript:

The tensile strength of TCFs and RCFs are 0.27±0.02 GPa and 1.33±0.08 GPa, respectively, and their elastic moduli are 10.45±1.24 GPa and 37.45±7.47 GPa, respectively (Fig. 4g). RCFs have a denser structure and a more oriented structure compared to TCFs, which results in higher friction and more difficult slippage between CNTs in the fibers, thus achieving significantly better mechanical properties. The elongation at break of TCFs and RCFs are 6.12±0.43% and 5.62±0.18%, respectively. Although TCFs and RCFs have similar elongation at break, RCFs exhibit significantly higher fracture work due to their significantly higher mechanical strength. Note that the fracture work of the TCFs and RCFs are 10.89±1.13 MJ m⁻³ and 47.54±3.85 MJ m⁻³, respectively (Fig. 4h).

Page 17 in Manuscript:

We also proved that our CNT fibers have high electrical conductivity, and the electrical conductivity of the CNT fibers with a density of 1.49 g cm⁻³ is as high as (6.03±0.25)×10⁵ S m⁻¹, which is similar to that of the CNT fibers with similar structures prepared by the similar method⁶⁷.

Comment 6: *Minor comment, in Table S1, it is recommended to change “Layer number” to “CNT type”. “MWNT” and “SWNT” are not numerical values.*

Response: We thank the reviewer for the comments.

Per the suggestion of the reviewer, we have changed “Layer number” to “CNT type” in Supplementary Table 1 in the revised manuscript.

Corresponding changes:

Page 3 in Supplementary Information:

Supplementary Table 1. Comparison of the structures and morphologies of our prepared CNTs with those of other reported CNTs.

Biomass	Continuous preparation	Macroscopic morphology	Synthesis temperature (°C)	I_G/I_D	CNT type	Micro morphology	Reference
Lignin	Yes (120 m h ⁻¹)	CNT fiber	1400	3.84	MWNTs	CNT aerogels	Our work
Eucalyptus oil	No	powder	850	3.3	SWNTs/ MWNTs	Carbon/CNT mixture	1
Turpentine oil	No	powder	700	0.93	MWNTs	CNT arrays	2
Sesame oil	No	powder	900	0.98	MWNTs	CNT arrays	3
Grass	No	powder	600	2.0	MWNTs	CNT arrays	4
Plant	No	powder	> 600	-	MWNTs	Carbon/CNT mixture	5
Poplar leaves	No	powder	450	0.79	MWNTs	Carbon/CNT mixture	6
Lignin	No	powder	2000	1.0	MWNTs	Carbon/CNT mixture	7

Note: SWNTs: single wall carbon nanotubes; MWNTs: multiwalled carbon nanotubes.

REVIEWER COMMENTS

Reviewer #1 (Remarks to the Author):

All issues have been addressed properly. Therefore, I recommend this paper for publication.

Reviewer #2 (Remarks to the Author):

Title: Continuously Processing Waste Lignin into High-value Carbon Nanotube Fibers

Manuscript#: NCOMMS-22-03434B

Comments : After careful reading, I think that the authors have made a good revision of the article based on the comments of the reviewers. This revised manuscript can be accepted and published by Nature Communications.

Reviewer #3 (Remarks to the Author):

Thank you for the correction made. However, the manuscript still contains several critical errors because of which it cannot be published by Nature Communications.

- 1) Regardless of what the authors claim, nanocarbon material having ID/IG of about 0.25 cannot be called "well crystallized". Judging on the TGA thermogram shown in Fig. 3h, there is significant weight loss between 300 and 400C, showing the removal of plenty of functional groups and defects.
- 2) Electrical conductivity was measured with a two-probe approach, which is inappropriate for materials of high electrical conductivity.
- 3) Calculation of electrical conductivity is also puzzling. In Fig. R3 in the SEM micrograph, the authors indicate thickness, while the material does not appear flat. If it is flat, how can it be called fiber?
- 4) Another problem with this article is that the lignin-derived material does not show benefits compared to CNTs made from synthetic precursors in terms of properties.
- 5) Last but not least, the most crucial issue of this submission is that there is a lack of convincing proof that the CNTs come from lignin. Lignin is injected into the furnace in methanol, which can also be used for making CNT fibers.

Point-by-point Response Letter

We would like to thank the three reviewers for their constructive comments. We have carried out additional experiments and discussion to address the reviewers' comments point-by-point. Please find our detailed responses in the response letter.

Reviewer #1

Comments: *All issues have been addressed properly. Therefore, I recommend this paper for publication.*

Response: We thank the reviewer for the positive comments.

Reviewer #2

Title: Continuously Processing Waste Lignin into High-value Carbon Nanotube Fibers

Manuscript#: NCOMMS-22-03434B

Comments: *After careful reading, I think that the authors have made a good revision of the article based on the comments of the reviewers. This revised manuscript can be accepted and published by Nature Communications.*

Response: We thank the reviewer for the positive comments.

Reviewer #3

Thank you for the correction made. However, the manuscript still contains several critical errors because of which it cannot be published by Nature Communications.

Response: We thank the reviewer for the comments.

Comment 1: *Regardless of what the authors claim, nanocarbon material having I_D/I_G of about 0.25 cannot be called "well crystallized". Judging on the TGA thermogram shown in Fig. 3h, there is significant weight loss between 300 and 400°C, showing the removal of plenty of functional groups and defects.*

Response: We thank the reviewer for the comments.

In our work, the CNTs synthesized from lignin are multi-walled (Figs. 3e and 3f). According to the Raman result, our CNTs exhibit an I_G/I_D value of 3.84 (I_D/I_G is about 0.25) (Fig. 3g), which is higher than or similar to that of multi-walled CNTs (MWNTs) prepared from fine chemicals and other biomass (Supplementary Table 1). The I_G/I_D values of MWNTs are generally lower than those of single-walled CNTs (SWNTs) and double-walled CNTs (DWNTs) due to the edge unsaturated carbon atoms, asymmetric carbon atoms and sidewall structural defects in the MWNTs^{1,2}. These functional groups and defects will be removed at high temperatures (300-400°C), thus exhibiting mass loss of approximately 2.3% in TGA result (Fig. 3h).

Fig. 3 Synthesis and structures of the lignin-derived CNTs. (a) Digital image and (b) SEM image of a CNT sock. (c) SEM image and (d) diameter distribution of the CNTs. (e, f) TEM images of the CNTs. (g) Raman spectrum and (h) thermogravimetric analysis (TGA) of the CNTs. (i) TEM image of the CNTs and catalyst particles. (j) Lattices of the CNTs and iron

catalysts. (k) Schematic showing the growth mechanism of the CNTs on iron catalysts. (l) XRD pattern of the CNTs. (m) Elemental mapping images of the CNTs. (n) TEM-EDS image of the CNTs and the element contents.

Corresponding changes:

Page 9 in Manuscript:

The CNTs have an I_G/I_D value of 3.84 (Fig. 3g), which is higher than or similar to that of multi-walled CNTs (MWNTs) prepared from fine chemicals and other biomass (Supplementary Table 1). The I_G/I_D values of MWNTs are generally lower than those of single-walled CNTs (SWNTs) and double-walled CNTs (DWNTs) due to the edge unsaturated carbon atoms, asymmetric carbon atoms and sidewall structural defects in the MWNTs^{45,46}. These functional groups and defects will be removed at high temperatures (300-400°C), thus exhibiting mass loss of approximately 2.3% in TGA result (Fig. 3h).

Page 3 in Supplementary Information:

Supplementary Table 1. Comparison of the structures and morphologies of our prepared CNTs with those of other reported CNTs.

Biomass	Carbon source type	Continuous preparation of CNT fiber	Synthesis temperature (°C)	I_G/I_D	CNT type	Micro morphology	Reference
Lignin	Biomass	Yes (120 m h ⁻¹)	1400	3.84	MWNTs	CNT aerogels	Our work
Eucalyptus oil	Biomass	No	850	3.3	SWNTs/ MWNTs	CNT powders	1
Turpentine oil	Biomass	No	700	0.93	MWNTs	CNT arrays	2
Sesame oil	Biomass	No	900	0.98	MWNTs	CNT arrays	3
Grass	Biomass	No	600	2.0	MWNTs	CNT arrays	4
Plant	Biomass	No	> 600	-	MWNTs	CNT powders	5
Poplar leaves	Biomass	No	450	0.79	MWNTs	CNT powders	6
Methane	Chemicals	Yes	1200	1.2	MWNTs	CNT aerogels	7
Hexane	Chemicals	Yes	1150-1500	3.36- 10.43	MWNTs	CNT aerogels	8
Ethanol	Chemicals	Yes	1150–1300	3.74	MWNTs	CNT aerogels	9
Toluene	Chemicals	Yes	1200	1.78- 14.28	MWNTs	CNT aerogels	10

Acetylene	Chemicals	Yes	-	1.22	MWNTs	CNT arrays	11
Toluene	Chemicals	Yes	1150	0.94	MWNTs	CNT aerogels	12
Xylene/dichlorobenzene	Chemicals	No	800	1.11	MWNTs	CNT arrays	13
Methane	Chemicals	Yes	1175	3.61	MWNTs	CNT aerogels	14
Methane	Chemicals	Yes	1200	7.7-14.3	SWNTs/DWNTs	CNT aerogels	7
Toluene	Chemicals	Yes	1200	1.92-33.33	SWNTs	CNT aerogels	10
Acetylene/Ethylene	Chemicals	Yes	1175	18.52-69.87	SWNTs/DWNTs	CNT aerogels	14
Ethanol	Chemicals	Yes	1000	> 30	SWNTs	CNT aerogels	15
Carbon monoxide	Chemicals	Yes	800-1050	10-440	SWNTs	CNT films	16
Acetone	Chemicals	Yes	1200	10-78.69	SWNTs	CNT aerogels	17
Methane	Chemicals	Yes	1200	> 60	SWNTs/DWNTs	CNT aerogels	18
Carbon monoxide	Chemicals	Yes	880	224	SWNTs	CNT films	19

Note: SWNTs: single-walled carbon nanotubes; DWNTs: double-walled carbon nanotubes; MWNTs: multi-walled carbon nanotubes.

Reference:

1. Hiura, H., Ebbesen, T., Fujita, J., Tanigaki, K. & Takada, T. Role of sp³ defect structures in graphite and carbon nanotubes. *Nature* **367**, 148-151 (1994).
2. Tsetseris, L. & Pantelides, S. Defect formation and hysteretic inter-tube displacement in multi-wall carbon nanotubes. *Carbon* **49**, 581-586 (2011).

Comment 2: *Electrical conductivity was measured with a two-probe approach, which is inappropriate for materials of high electrical conductivity.*

Response: We thank the reviewer for the comments.

Per the suggestion of the reviewer, we remeasured the electrical conductivity of the lignin-derived CNT fibers using a four-probe tester (RTS-9, Guangzhou Four Probe Technology Co., Ltd., China) (Supplementary Fig. 25). The electrical conductivity (σ , S m⁻¹) was calculated by the following equation: $\sigma=L/(RS)$, where L is the distance between the four probes (L=1 mm),

R is the resistance of the CNT fibers (Ω), and S is the cross-sectional area (m^2). RCFs with a rectangular cross-section were used for the electrical conductivity determination, and 10 samples were determined to obtain an average value. The result of four-probe determination shows that our CNTs have an electrical conductivity of $(1.19 \pm 0.09) \times 10^5 \text{ S m}^{-1}$.

Supplementary Figure 25. Electrical conductivity test of the CNT fibers using four-probe method. (a) Digital image showing the electrical conductivity measurement by four-probe method. (b) SEM image of the CNT fibers to show the thickness (t). (c, d) SEM images of the CNT fibers to show the width (w).

Corresponding changes:

Page 17 in Manuscript:

We also proved that our CNT fibers have high electrical conductivity, and the electrical conductivity of the CNT fibers with a density of 1.49 g cm^{-3} is as high as $(1.19 \pm 0.09) \times 10^5 \text{ S m}^{-1}$ (Supplementary Fig. 25), which is similar to that of the CNT fibers with similar structures prepared by the similar method⁷⁰.

Page 23 in Manuscript:

The electrical conductivity of the CNT fibers was measured using a four-probe tester (RTS-9, Guangzhou Four Probe Technology Co., Ltd., China). The electrical conductivity (σ , S m^{-1}) was calculated by the following equation: $\sigma=L/(RS)$, where L is the distance between the four probes ($L=1 \text{ mm}$), R is the resistance of the CNT fibers (Ω), and S is the cross-sectional area

(m²). RCFs with a rectangular cross-section were used for the electrical conductivity determination, and 10 samples were determined to obtain an average value.

Comment 3: *Calculation of electrical conductivity is also puzzling. In Fig. R3 in the SEM micrograph, the authors indicate thickness, while the material does not appear flat. If it is flat, how can it be called fiber?*

Response: We thank the reviewer for the comments.

We remeasured the electrical conductivity of the CNT fibers using a four-probe tester (RTS-9, Guangzhou Four Probe Technology Co., Ltd., China) (Supplementary Fig. 25). RCFs with a rectangular cross-section were used for the electrical conductivity determination, and 10 new samples were determined to obtain the average electrical conductivity.

According to the definition of fiber^{1,2,3,4}, fibers (or “fibres” outside the United States) are discrete “slender” objects able to transmit tensile but not compressive axial loads. They have a relatively high aspect or length-to-diameter ratio (not universally agreed but certainly $\gg 100$). Because the material we obtained is a continuous filament, it conforms to the fiber definition and can be called a fiber regardless of the shape of its cross-section. The cross-section shapes of our CNT fibers obtained by different densification methods are different. The cross-section of the CNT fibers obtained by twisting process is circular, while the cross-section of the CNT fibers obtained by rolling method is rectangular.

Because the CNT fibers obtained by rolling method have a more densified structure, resulting in better mechanical properties, electrical conductivity and thermal conductivity than the CNT fibers obtained by twisting. Therefore, we mainly studied the properties of the CNT fibers obtained by rolling method in our work.

Supplementary Figure 25. Electrical conductivity test of the CNT fibers using four-probe method. (a) Digital image showing the electrical conductivity measurement by four-probe method. (b) SEM image of the CNT fibers to show the thickness (t). (c, d) SEM images of the CNT fibers to show the width (w).

Corresponding changes:

Page 23 in Manuscript:

The electrical conductivity of the CNT fibers was measured using a four-probe tester (RTS-9, Guangzhou Four Probe Technology Co., Ltd., China). The electrical conductivity (σ , S m⁻¹) was calculated by the following equation: $\sigma=L/(RS)$, where L is the distance between the four probes (L=1 mm), R is the resistance of the CNT fibers (Ω), and S is the cross-sectional area (m²). RCFs with a rectangular cross-section were used for the electrical conductivity determination, and 10 samples were determined to obtain an average value.

References:

1. Murthy, H. S. Introduction to textile fibres. (CRC Press, Florida, 2016).
2. Sinclair, R. Understanding textile fibres and their properties: what is a textile fibre? In: Textiles and fashion (Elsevier, Amsterdam, 2015).
3. Hawkins, S. C. Nanotube superfiber materials: Chapter 1. Introduction to fiber materials (Elsevier, Amsterdam, 2013).
4. Hearle, J. Fibre structure: its formation and relation to performance. In: Handbook of

textile fibre structure (Elsevier, Amsterdam, 2009).

Comment 4: *Another problem with this article is that the lignin-derived material does not show benefits compared to CNTs made from synthetic precursors in terms of properties.*

Response: We thank the reviewer for the comments.

First of all, in terms of mechanical property, our CNT fibers exhibit a tensile strength of 1.33 GPa, which is higher than or similar to that of most reported biomass-derived carbon fibers, array CNT fibers, CNT fibers from FCCVD and wet-spun CNT fibers, as well as all common metal materials (Fig. 4j). It should be emphasized that the mechanical strength of our CNT fibers exceeds that of most CNT fibers prepared with fine chemicals (such as alkanes and aromatic hydrocarbons) as carbon sources ^{1,2,3}.

In terms of electrical conductivity, our CNT fibers have an electrical conductivity of $1.19 \times 10^5 \text{ S m}^{-1}$, which is similar to that of the CNT fibers with similar structures prepared by the similar method ⁴. The electrical conductivity of our CNT fibers is higher than that of almost all reported biomass-derived carbon fibers and array CNT fibers as well as most commercial carbon fibers (Fig. 4j). The electrical conductivity of our CNT fibers is lower than that of the wet-spun CNT fibers, which may be due to the higher purity and crystallinity of the CNTs used for wet-spinning as well as the higher density of the resultant CNT fibers (Fig. 4j).

In addition to excellent mechanical and electrical properties, our CNT films with a density of 0.82 g cm^{-3} exhibit high thermal conductivity of $33.21 \text{ W m}^{-1} \text{ K}^{-1}$. Compared to biomass-derived carbon materials ($0.06\text{-}24 \text{ W m}^{-1} \text{ K}^{-1}$), our CNT films possess higher thermal conductivity, comparable to that of the CNT films with similar characteristics prepared by the similar method ($20.91\text{-}458.58 \text{ W m}^{-1} \text{ K}^{-1}$) as well as some common metals ($30\text{-}500 \text{ W m}^{-1} \text{ K}^{-1}$) (Fig. 4i). It should be emphasized that the thermal conductivity of CNT films increases gradually with increasing density ⁵. Considering that our CNT films have significantly lower density (0.82 g cm^{-3}) than common metals ($2.7\text{-}10.49 \text{ g cm}^{-3}$), they can be used in some fields that require lightweight thermal conductive materials.

In addition to material properties, cost and energy consumption also need to be considered. Lignin was used as the carbon source to prepare CNT fibers in our work, and it is derived from the by-products of the pulp and paper industry, and the cost is negligible. From the point of

view of energy consumption, the energy consumption of our CNT fibers is estimated to be about 0.12 MJ m^{-1} , which is significantly lower than that of lignin-based carbon fibers prepared by traditional methods ($0.22\text{-}0.67 \text{ MJ m}^{-1}$) (Supplementary Fig. 26).

Fig. 4 Preparation and properties of lignin-derived CNT fibers. Preparation diagrams of (a) TCFs and (b) RCFs. SEM images of (c) TCFs and (d) RCFs. Polarised Raman spectra of (e) TCFs and (f) RCFs. (g) Tensile stress-strain curves of the TCFs and RCFs. (h) Comparison of tensile strength and fracture work for the TCFs and RCFs. Error bars represent s.d. ($n=5$). (i) Comparison of the thermal conductivity and density between our CNT films and other conductive materials^{48, 56, 57, 58, 59, 60, 61, 62, 63}. (j) Comparison of the tensile strength and electrical

conductivity of our CNT fibers with commercial carbon fibers⁶⁴, wet-spun CNT fibers^{18, 65, 66, 67}, CNT fibers from FCCVD^{25, 68, 69, 70, 71}, CNT fibers from array spinning^{72, 73, 74, 75, 76}, common metal materials⁶³ and biomass-derived carbon fibers^{77, 78, 79, 80, 81, 82, 83}.

Last but not least, we need to reemphasize that our work achieved for the first time the continuous preparation of CNT fibers from waste lignin as carbon source. The preparation rate of the CNT fibers is up to 120 m h⁻¹, which is significantly higher than that of lignin-based carbon fibers prepared by traditional methods (20-35 m h⁻¹) (Supplementary Fig. 26). The high preparation efficiency and low energy consumption combined with low lignin pretreatment requirements make our method very promising for large-scale production of lignin-based CNT fibers.

Supplementary Figure 26. Comparison of energy consumption between lignin-based CNT fibers prepared by FCCVD method in our work and lignin-based carbon fibers prepared by conventional spinning method^{63,64}.

References:

1. Watanabe, T. et al. Post-synthesis treatment improves the electrical properties of dry-spun carbon nanotube yarns. *Carbon* **185**, 314-323 (2021).
2. Tran, T. Q. et al. Purification and dissolution of carbon nanotube fibers spun from the

- floating catalyst method. *ACS Appl. Mater. Interfaces* **9**, 37112-37119 (2017).
3. Zhu, H., Xu, C., Wu, D., Wei, B., Vajtai, R. & Ajayan, P. Direct synthesis of long single-walled carbon nanotube strands. *Science* **296**, 884-886 (2002).
 4. Wang, J. N., Luo, X. G., Wu, T. & Chen, Y. High-strength carbon nanotube fibre-like ribbon with high ductility and high electrical conductivity. *Nat. Commun.* **5**, 3848 (2014).
 5. Zhan, H., Chen, Y. W., Shi, Q. Q., Zhang, Y., Mo, R. W. & Wang, J. N. Highly aligned and densified carbon nanotube films with superior thermal conductivity and mechanical strength. *Carbon* **186**, 205-214 (2022).

Comment 5: *Last but not least, the most crucial issue of this submission is that there is a lack of convincing proof that the CNTs come from lignin. Lignin is injected into the furnace in methanol, which can also be used for making CNT fibers.*

Response: We thank the reviewer for the comments.

In order to demonstrate that the CNTs were derived from lignin rather than methanol, additional experiments were supplemented. We prepared the lignin solutions with lignin concentrations of 5.5 mg mL⁻¹, 2.5 mg mL⁻¹, 1.6 mg mL⁻¹, 0.8 mg mL⁻¹, 0.4 mg mL⁻¹ and 0 mg mL⁻¹, respectively using methanol as solvent. With the decrease of lignin concentration, the amount of product produced from the tubular furnace decreased gradually (Supplementary Fig. 17). When the lignin concentration decreased to 0.8 mg mL⁻¹, CNT fibers cannot be prepared continuously (Supplementary Fig. 17j). When the lignin concentration is lower than 0.4 mg mL⁻¹, only a very small amount of product can be formed (Supplementary Fig. 17k). For the solution without lignin, the product cannot be observed at the outlet and inside of the tubular furnace (Supplementary Fig. 17i). These results indicated that our CNTs were synthesized from lignin, and pure methanol cannot be used as a carbon source to prepare CNTs, which is in line with some reported work^{1,2}.

Supplementary Figure 17. Effect of lignin concentration on the synthesis of CNTs. Digital images of lignin solutions with concentrations of (a) 5.5 mg mL^{-1} , (b) 2.5 mg mL^{-1} , (c) 1.6 mg mL^{-1} , (d) 0.8 mg mL^{-1} , (e) 0.4 mg mL^{-1} , (f) 0 mg mL^{-1} . Digital images showing the synthesis of CNTs from lignin solutions with concentrations of (g) 5.5 mg mL^{-1} , (h) 2.5 mg mL^{-1} , (i) 1.6 mg mL^{-1} , (j) 0.8 mg mL^{-1} , (k) 0.4 mg mL^{-1} , (l) 0 mg mL^{-1} .

Corresponding changes:

Page 12 in Manuscript:

The effect of lignin concentration on the synthesis of CNTs was investigated. With the decrease of lignin concentration, the amount of product produced from the tubular furnace decreased gradually (Supplementary Fig. 17). When the lignin concentration decreased to 0.8 mg mL^{-1} , CNT fibers cannot be prepared continuously (Supplementary Fig. 17j). When the lignin concentration is lower than 0.4 mg mL^{-1} , only a very small amount of product can be formed (Supplementary Fig. 17k). For the solution without lignin, the product cannot be observed at the outlet and inside of the tubular furnace (Supplementary Fig. 17i). These results indicated that the CNTs were synthesized from lignin, and pure methanol cannot be used as a carbon source to prepare CNTs, which is in line with some reported work^{53,54}.

References:

1. Wu, Z. P., Wang, J. N. & Ma, J. Methanol-mediated growth of carbon nanotubes. *Carbon* **47**, 324-327 (2009).
2. Li, Y., Zhang, L., Zhong, X. & Windle, A. H. Synthesis of high purity single-walled carbon nanotubes from ethanol by catalytic gas flow CVD reactions. *Nanotechnology* **18**, 225604 (2007).

REVIEWERS' COMMENTS

Reviewer #3 (Remarks to the Author):

The provided explanation is convincing, so the manuscript can be accepted for publication.